# VisionThink: Smart and Efficient Vision Language Model via Reinforcement Learning

**Senqiao Yang**[* 1,3]   **Junyi Li**[* 2,3]   **Xin Lai**[* 3]   **Jinming Wu**[3]
**Wei Li**[3]   **Bei Yu**[1]   **Hengshuang Zhao**[‡2]   **Jiaya Jia**[1,4]

[1]CUHK        [2]HKU        [3]ByteDance        [4]HKUST

Codes and models: `https://github.com/dvlab-research/VisionThink`

## Abstract

Recent advancements in vision-language models (VLMs) have improved performance by increasing the number of visual tokens, which are often significantly longer than text tokens. However, we observe that most real-world scenarios do not require such an extensive number of visual tokens. While the performance drops significantly in a small subset of OCR-related tasks, models still perform accurately in most other general VQA tasks with only 1/4 resolution. Therefore, we propose to dynamically process distinct samples with different resolutions, and present a new paradigm for visual token reduction, namely, VisionThink. It starts with a downsampled image and smartly decides whether it is sufficient for problem solving. Otherwise, the model could output a special token to request the higher-resolution image. Compared to existing Efficient VLM methods that reduce tokens using fixed pruning ratios or thresholds, VisionThink autonomously decides whether to reduce tokens case by case. As a result, it demonstrates strong fine-grained visual understanding capability on OCR-related tasks, and meanwhile saves substantial visual tokens on simpler tasks. We adopt reinforcement learning and propose the LLM-as-Judge strategy to successfully apply RL to general VQA tasks. Moreover, we carefully design a reward function and penalty mechanism to achieve a stable and reasonable image resize call ratio. Extensive experiments demonstrate the superiority, efficiency, and effectiveness of our method.

## 1 Introduction

Recently, Vision-Language Models (VLMs) [31, 30, 33, 9, 3] have achieved remarkable performance in general visual question answering (General VQA) and various real-world scenarios by projecting and adapting visual tokens into the LLM space [66, 1, 102, 4]. However, as the performance of VLMs continues to advance, the consumption of visual tokens has grown exponentially. For instance, a 2048×1024 image captured by a smartphone requires 2,678 visual tokens in Qwen2.5-VL [5], which significantly exceeds the number of text tokens. This leads to substantial memory consumption and notable latency, further constraining the deployment of VLMs on edge devices. Therefore, it is imperative to minimize the excessive use of visual tokens.

Numerous works on visual token reduction have been proposed [71, 59, 18, 22, 8, 100, 80]. Most approaches prune or merge a fixed number of visual tokens using predetermined thresholds. However, redundancy levels vary across different questions and images, leading to a natural question: *Should we really apply a uniform token reduction ratio across all scenarios?*

To answer this question, we simply reduced the image resolution to decrease the number of visual tokens and evaluated Qwen2.5-VL's[5] performance on several benchmarks. As shown in the left of Fig. 1, we found that for most real-world scenarios (general VQA scenarios), such as MME

---

[*]Equal Contribution, [‡]Corresponding author

39th Conference on Neural Information Processing Systems (NeurIPS 2025).

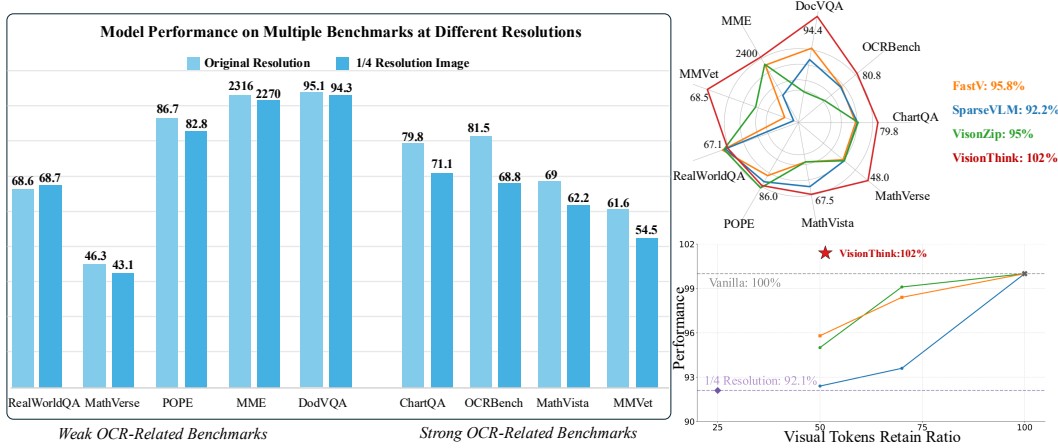

Figure 1: **Our key observations and VisionThink performance and efficiency**. **Left**: We find that in most general scenarios, even reducing visual tokens by a factor of four results in only minimal performance drop. However, token reduction leads to a significant performance drop on strong OCR-related benchmarks. **Right**: Our VisionThink significantly outperforms previous work in both performance and efficiency.

and RealWorldQA, even reducing the image resolution by a factor of four, which significantly cuts visual tokens by 75%, has minimal impact on the model's performance. However, as shown in the right of Fig. 1, for OCR-realted scenarios such as ChartQA and OCRBench, which require detailed understanding and OCR-related capabilities, reducing the number of visual tokens leads to a significant drop in performance. Based on these observations, we find that most real-world questions do not require high-resolution images with long visual tokens, while a small subset of OCR-related tasks demand such detailed input much. And a uniform token reduction ratio should not be applied across all tasks. Therefore, there is significant potential for efficiency optimization if we can dynamically distinguish between samples that require high-resolution processing and those that do not.

In this paper, we propose VisionThink, a new EfficientVLM paradigm that leverages the model's reasoning capabilities. Unlike prior methods that process full images and later discard redundant tokens, VisionThink directly inputs reduced visual tokens and allows the model to request the original high-resolution image when needed. This enables more efficient inference in most real-world scenarios, and meanwhile preserving performance on OCR-related tasks.

Although VisionThink offers a promising way to handle samples with varying levels of visual redundancy smartly, it still faces two key challenges:

**Effective Reinforcement Learning for General VQA.** Conventional rule-based reinforcement learning algorithms, typically used to optimize reasoning process, struggle with the diversity and complexity of general VQA. To overcome this issue, we propose the LLM-as-Judge approach, enabling semantic matching. Experiments show performance improvement across several general VQA benchmarks, highlighting the potential to extend vision-based reinforcement learning beyond visual math reasoning to broader VQA tasks.

**Determine When High Resolution is Worth.** To improve efficiency without compromising performance, the model must accurately determine when high-resolution input is necessary. We achieve this by carefully designing a balanced reward function to prevent the model from collapsing into always requiring high-resolution images or always using low-resolution images. With this mechanism, VisionThink maintains strong performance on OCR benchmarks while delivering significant speed-ups on non-OCR benchmarks, achieving up to 100% for DocVQA.

Overall, we present a simple yet effective pipeline—VisionThink. It introduces a new approach to visual token reduction by dynamically determining reduction based on the content of each sample, thereby achieving efficiency gains at the sample level. Consequently, it is compatible with other advanced spatial-level methods. We hope our work sheds new light on this area.

## 2 Preliminary

### 2.1 Large Language Models and Reinforcement Learning

Recent progress in improving the reasoning ability of large language models (LLMs)[16, 21] has shown that Reinforcement Learning (RL) is an effective training approach. In this work, we use Group Relative Policy Optimization (GRPO)[56] as our training method. GRPO removes the need for a separate critic model by using group scores to estimate baselines. This reduces computation cost, improves training stability, and leads to faster and more reliable performance gains.

During the training process, GRPO samples a group of outputs $\{o_1, o_2, \cdots, o_G\}$ based on the given question $q$ from the old policy $\pi_{\theta_{old}}$ and then optimizes the policy model $\pi_\theta$ by maximizing the following objective:

$$\mathcal{J}_{GRPO}(\theta) = \mathbb{E}_{[q \sim \mathcal{D}, \{o_i\}_{i=1}^G \sim \pi_{\theta_{old}}(\cdot|q)]}$$

$$\frac{1}{G} \sum_{i=1}^G \left( \min \left( \frac{\pi_\theta(o_i|q)}{\pi_{\theta_{old}}(o_i|q)} A_i, \text{clip} \left( \frac{\pi_\theta(o_i|q)}{\pi_{\theta_{old}}(o_i|q)}, 1 - \epsilon, 1 + \epsilon \right) A_i \right) - \beta \mathbb{D}_{KL} \left( \pi_\theta || \pi_{ref} \right) \right) \tag{1}$$

$$\mathbb{D}_{KL} \left( \pi_\theta || \pi_{ref} \right) = \frac{\pi_{ref}(o_i|q)}{\pi_\theta(o_i|q)} - \log \frac{\pi_{ref}(o_i|q)}{\pi_\theta(o_i|q)} - 1, \tag{2}$$

where $q$ represents the input questions drawn from the dataset $\mathcal{D}$, and $o$ denotes the generated text response. $\mathbb{D}_{KL}$ is the KL-divergence measure, while $\epsilon$ and $\beta$ are hyper-parameters. $A_i$ indicates the advantage, computed using a group of rewards $\{r_1, r_2, \ldots, r_G\}$ corresponding to the outputs within each group.

### 2.2 Computation Complexity

To evaluate the computational complexity of VLMs, we analyze key components, including the self-attention mechanism and the feedforward network (FFN). The total floating-point operations (FLOPs) are given by:

$$\text{Total FLOPs} = T \times (4nd^2 + 2n^2d + 2ndm)$$

where $T$ denotes the number of transformer layers, $n$ is the sequence length, $d$ is the size of the hidden dimension, and $m$ is the intermediate size of the FFN.

This equation indicates that computational complexity is largely determined by the sequence length $n$. In general VLM tasks, the total sequence length can be expressed as $n = n_{sys} + n_{img} + n_{question}$, where $n_{img}$—the number of image tokens—is typically much larger than the other two components, often reaching hundreds or even thousands. As a result, the prefilling stage dominates the total inference time in most VLM scenarios.Hence, controlling the number of image tokens is key to achieving VLM efficiency.

## 3 Methodology

### 3.1 Overview

Our objective is to develop a smart and efficient VLM, capable of autonomously determining whether the information in the given image is sufficient to answer the question accurately. As shown in Fig. 2, the pipeline first processes a low-resolution image to minimize the computataion cost. It then smartly requests original high-resolution inputs when the information in the downsampled image is insufficient to answer the question. Ideally, this strategy maintains high performance while sharply reducing computational load. To achieve this goal, we must address two key challenges:

**Effective RL on General VQA.** Due to the diversity and complexity of general VQA, traditional rule-based RL algorithms are not directly applicable. To address this, we propose an LLM-as-Judge strategy, in which a large language model guides and evaluates the RL training process (Sec. 3.2). We further extend the Multi-Turn GRPO algorithm to suit our setting (Sec. 3.3).

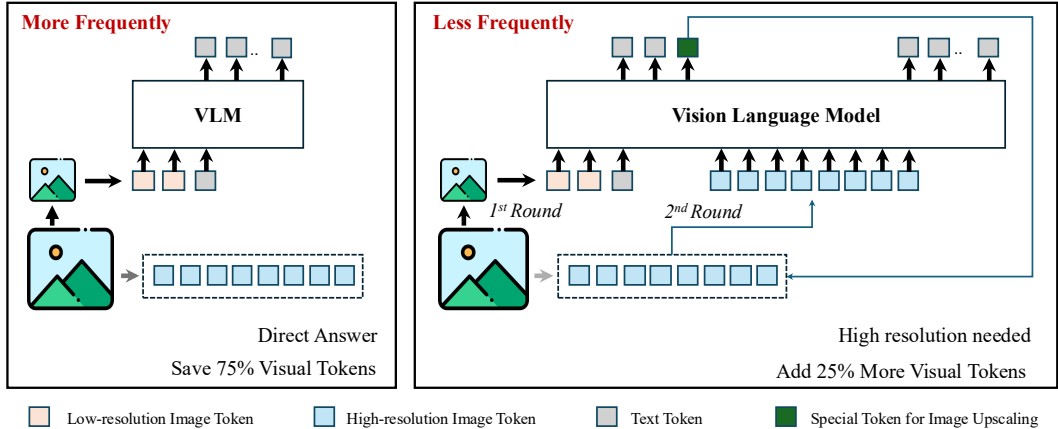

Figure 2: **Framework of VisionThink.** (a) The left image illustrates VisionThink processing an image with resolution reduced by a factor of four, where the VLM directly provides an answer. (b) The right image shows a case where the model detects insufficient information and requests a high-resolution image to answer the question.

**Enabling the model to decide when high resolution is necessary.** The model must learn to assess whether a downsampled image contains sufficient information to answer the question if the original high-resolution image is required. So that the model could balance the efficiency and performance. To this end, we design a reward function that encourages optimal resolution decisions (Sec. 3.4) and collect training data across multiple resolutions to support effective learning (Sec. 3.5).

## 3.2    LLM-as-Judge for General VQA

**Challenges.** One of the central challenges in applying reinforcement learning to General VQA lies in evaluating model responses, especially when answers are open-ended or context-dependent. Most existing multi-modal RL efforts remain limited to structured tasks such as visual math, where ground-truth answers can be easily defined and verified via rules or exact matching. However, this approach breaks down in General VQA settings, where the diversity and ambiguity of valid answers make rule-based verification infeasible.

**Pure Text Accuracy Judgement.** To address this, we employ an external LLM as a judgment evaluator. Leveraging its broad knowledge and language understanding, the LLM assesses the correctness of model outputs in a human-aligned and flexible manner. Importantly, the evaluation is conducted purely in text by comparing the model's answer with the ground-truth. This design avoids biases from visual content and the limitations of VLM performance. Furthermore, to minimize potential misjudgment by the evaluator, the reward is discrete (either 0 or 1) rather than continuous. The detailed judgment prompt is shown in Appendix B.1.

**Effectiveness.** The LLM-as-Judge is flexible, one advantage is that most of the SFT data could be used. To verify the effectiveness of our proposed LLM-as-Judge, we collected 130K samples (filtered from the open-sourced datasets), which can be directly used to train the model with GRPO, without requiring any cold-start process. The results show significant improvement compared to the base model, Qwen2.5VL-Instruct. Further details are provided in Appendix B.5.

## 3.3    Mutli-Turn Training Algorithm

**Multi-Turn GRPO.** In our VisionThink framework, we first input the question and the downsampled image into the VLM. If the information is insufficient to answer the current question, the model will autonomously request a higher-resolution image and generate a new response. This process is essentially a multi-turn interaction. Therefore, we extend the original GRPO (Eq. 1) to a multi-turn GRPO, as shown in Eq. 3:

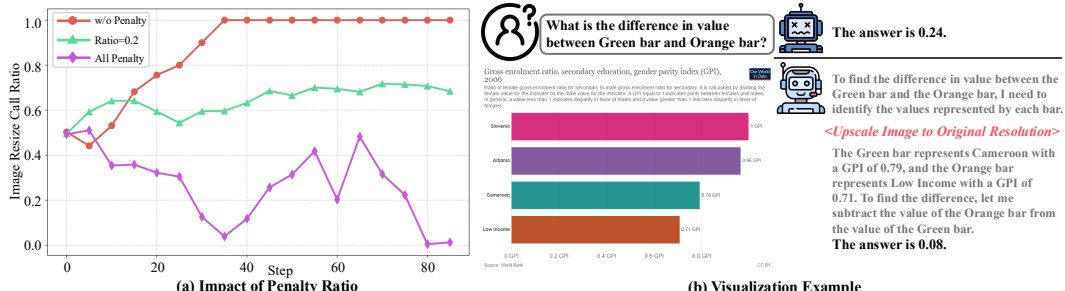

Figure 3: (a) Impact of the Penalty Ratio. Applying a penalty to all resize image requests or removing the penalty entirely will both lead to model collapse. (b) VisionThink correctly solves OCR-related problems by autonomously requesting high-resolution images.

$$\mathcal{J}_{GRPO}(\theta) = \mathbb{E}_{q \sim \mathcal{D}, \{o_i\}_{i=1}^{G} \sim \pi_{\text{old}}(\cdot|q;\mathcal{I})} \left[ \frac{1}{G} \sum_{i=1}^{G} \frac{1}{\sum_{t=1}^{|o_i|} \mathbb{I}(o_{i,t})} \sum_{t=1}^{|o_i|} \mathbb{I}(o_{i,t}) \right.$$

$$\left. \cdot \min \left( p_{i,t} \hat{A}_{i,t}, \text{clip}\left( p_{i,t}, 1 - \epsilon, 1 + \epsilon \right) \hat{A}_{i,t} \right) - \beta \mathbb{D}_{KL} \left[ \pi_\theta || \pi_{\text{ref}} \right] \right], \tag{3}$$

where $p_{i,t} = \frac{\pi_\theta(o_{i,t}|q,o_{i,<t};\mathcal{I})}{\pi_{\text{old}}(o_{i,t}|q,o_{i,<t};\mathcal{I})}$, and $\mathbb{I}(o_t)$ is the token loss masking operation such that $\mathbb{I}(o_t) = 1$ if $o_t$ is the generated token from LLM and $\mathbb{I}(o_t) = 0$ if $o_t$ is the response token from the called tools. Intuitively, we masked all text and image tokens from the user and performed optimization solely based on the multi-turn output tokens generated by the VLM.

**How does the model signal the need for a high-resolution image?** To determine when the model requires a high-resolution image, we modify the prompt to instruct the model to output specific special tokens. Notably, this is a non-trivial process because our training does not introduce any cold-start phase, which leads to a performance drop in general VQA (Appendix C.2). Therefore, selecting an appropriate and effective prompt at the early stage of training is crucial. The prompt must ensure that the model is capable of outputting the required special tokens during multi-turn rollouts in a zero-shot setting. Otherwise, GRPO will fail to optimize correctly due to the absence of gradients. We conduct comparative ablation studies in Appendix C.3 and find that the Agent Prompt recommended by Qwen-2.5VL [5] suits VisionThink best. The prompt details are provided in Appendix B.1.

### 3.4 Reward Design

Different reward functions can lead the model toward different optimization directions and final performance outcomes. The reward function in our VisionThink framework consists of three components:

$$\mathcal{R}_{\text{overall}} = \mathcal{R}_{\text{accuracy}} + \mathcal{R}_{\text{format}} - \mathcal{P}_{\text{control}}, \tag{4}$$

where $\mathcal{R}$ represents the reward and $\mathcal{P}$ represents the penalty.

**Accuracy Reward.** We utilize the LLM-as-Judge strategy to evaluate whether the generated answers are correct, where 0 denotes an incorrect answer and 1 denotes a correct one. The detailed design of the accuracy reward follows the description in Sec. 3.2.

**Format Reward.** To maintain the model's instruction-following capability and ensure that the trained model can more accurately call the image resize function, we apply a format reward. Specifically, we require the reasoning process to be enclosed in "<think></think>" tags, the final answer in "<answer></answer>" tags, and the function call to conform to the JSON format specified in Appendix B. If any of these formats are incorrect, the format score is 0. Only when all formats are correct can the model achieve the full format score of 0.5.

**Penalty Control.** The design of the penalty is a key component of the reward function. As shown in Fig. 3(a), since using high-resolution images generally improves performance, without any penalty, the model tends to collapse into always requesting high-resolution images. To prevent this, we initially

Table 1: **Effective Performance Compared to the Sota Model.** Our model is based on Qwen2.5-VL-7B-Instruct. VisionThink‡ represents a model trained on general VQA tasks using full image resolution with the LLM-as-Judge strategy, which does not contain efficiency capabilities. Qwen2.5-VL-7B* reports the results evaluate by lmms-eval[94].

| Method | MMMU | MMMU-Pro | MMBench | RealWorldQA | POPE | MME | MathVista | MathVerse | MMVet |
|---|---|---|---|---|---|---|---|---|---|
| | val | test | en_test | test | test | test | testmini | testmini | test |
| *Closed-Source Model* | | | | | | | | | |
| GPT-4o [50] | 69.1 | 54.0 | 83.4 | 58.6 | 85.6 | 2329 | 63.8 | 50.2 | 69.1 |
| Claude-3.5 Sonnet [2] | 68.3 | 55.0 | 82.6 | 59.9 | - | 1920 | 67.7 | 41.2 | 70.1 |
| Gemini-1.5-Pro [62] | 62.2 | 49.4 | 73.9 | 70.4 | 88.2 | - | 63.9 | - | 64.0 |
| *Open-Source General Model* | | | | | | | | | |
| Cambrain-1-8B [65] | 42.7 | - | 75.9 | 60.0 | 86.4 | 1803 | 49.0 | - | - |
| InternVL2-8B [12] | 49.3 | 32.5 | 81.7 | 64.4 | 84.2 | 2210 | 58.3 | - | 60.0 |
| LLaVA-OneVision-7B [28] | 48.8 | - | - | 66.3 | 88.4 | 1998 | 63.2 | - | 57.5 |
| MiniCPM-Llama-V-2.5-8B [89] | 45.8 | 19.6 | 77.2 | 63.0 | 86.7 | 2025 | 54.3 | - | - |
| MiniCPM-V-2.6-8B [89] | 49.8 | 27.2 | 78.0 | 65.0 | 83.2 | 2348 | 60.6 | - | - |
| IXC-2.5 [95] | 42.9 | - | 82.2 | 67.8 | - | 2229 | 63.8 | - | 51.7 |
| InternVL2.5-8B [11] | 56.0 | 38.2 | 84.6 | 70.1 | 90.6 | 2344 | 64.4 | 39.5 | 62.8 |
| *Reasoning Model* | | | | | | | | | |
| LLaVA-CoT-11B [78] | - | - | 75.0 | - | - | - | 54.8 | - | 60.3 |
| LLaVA-Reasoner-8B [97] | - | - | - | - | - | - | 50.6 | - | - |
| Insight-V-8B [14] | 50.2 | 24.9 | 82.3 | - | - | 2312 | 59.9 | - | - |
| Mulberry-7B [86] | 55.0 | - | - | - | - | 2396 | 63.1 | - | - |
| Vision-R1-LlamaV-CI-11B [19] | - | - | - | - | - | 2190 | 62.7 | 27.1 | - |
| *VisionThink* | | | | | | | | | |
| Qwen2.5-VL-7B* [5] | 50.3 | 37.7 | 82.6 | 68.6 | 86.7 | 2316 | 68.2 | 46.3 | 61.6 |
| VisionThink ‡ | 51.0 | 40.1 | 82.9 | 68.6 | 87.9 | 2307 | 71.2 | 48.8 | 67.5 |
| VisionThink | 51.2 | 38.9 | 80.0 | 68.5 | 86.0 | 2400 | 67.5 | 48.0 | 67.1 |

followed Search-R1 [23] and applied a 0.1 penalty for correct answers that relied on high-resolution images. However, this approach causes the model to favor direct answers, leading to a collapse where the model relies solely on direct answers, as indicated by the purple line in Fig. 3. The reason is that even blurry, low-resolution images sometimes allow the model to guess the correct answer, and the 0.1 penalty unintentionally reinforced this preference for direct answering.

To address this, we introduce a threshold to control the phenomenon of "lucky guesses". When the probability of correctly answering with a low-resolution image is low, we apply a 0.1 penalty to direct answers to encourage high-resolution requests; conversely, when the probability is high, we penalize high-resolution requests with a 0.1 penalty. In summary, the penalty is designed as below:

$$\mathcal{P}_{control} = 0.1 \cdot \left[ \mathbf{1}_{direct} \mathbb{I}(r < \theta) + \mathbf{1}_{high} \mathbb{I}(r \geq \theta) \right], \qquad r = \frac{C_{direct}}{C_{direct} + C_{high}}, \qquad (5)$$

where $C_{direct}$ and $C_{high}$ are the correct-answer counts for low- and high-resolution inputs, respectively, and $\mathbf{1}_{action}$ is the indicator of the chosen action, and we set $\theta$ as 0.2 here. We will discuss the impact of the threshold in Appendix C.3.

## 3.5 Data Preparation

To enable our model can decide when high resolution is necessary, we collect corresponding VQA samples, including both cases requiring high-resolution images and cases adequately answered using downsampled images. To achieve this, we use our base policy model, Qwen2.5VL-Instruct, to perform multiple rollouts on the training dataset and classify the samples based on accuracy. Specifically, we set the temperature to 1 and roll out each sample 8 times. If both the high-resolution and downsampled images yield correct answers in all 8 rollouts, we classify the sample as solvable

using low resolution. Conversely, if the number of correct answers using the high-resolution image exceeds that of the downsampled image by 6 or more, we classify the sample as requiring high resolution. By using the above method, we selected 10K samples that require high-resolution images and 10K samples that do not, to train our model.

## 4 Experiments

### 4.1 Evaluation Setup

**Benchmarks.** We evaluate VisionThink on several general VQA benchmarks, including ChartQA [45], OCRBench [37], MathVista [42], MMVet [91], RealWorldQA [74], and Math-Verse [96], etc. Notably, benchmarks such as ChartQA, OCRBench, and MathVista are strongly OCR-related, requiring the model to possess a high level of detail comprehension. The detailed descriptions of these benchmarks are shown in Appendix B.4.

**Implementation Details.** We conduct experiments based on Qwen2.5-VL-7B-Instruct[5]. For training, we employ veRL[58] framework and use a total batch size of $512$, with a mini-batch size of 32, we set the policy LLM learning rate to $1e - 6$ and sample 16 responses per prompt, ensuring a stable and effective training process. For inference, we use the vLLM framework and set the temperature to 0. Further details are shown in Appendix B.3.

### 4.2 Reinforcement Learning Enables VLM to Be More Effective

**Main Results.** To demonstrate the effectiveness of our VisionThink, we compare our VisionThink with the current open-source and closed-source state-of-the-art (sota) method. As shown in Table 6, VisionThink ‡ is used to demonstrate the effectiveness of the LLM-as-Judge strategy on general VQA tasks. It represents a model trained with full image resolution using only accuracy and format rewards, and thus does not incorporate efficiency capabilities. The results show that our VisionThink achieves comparable or even superior performance on general VQA tasks while being more efficient. Specifically, MathVerse and MMVet achieve scores of 48.0 and 67.1, representing improvements of 3.7% and 8.9%, respectively, over the base model. Furthermore, our model performs comparably to closed-source models on several benchmarks such as MathVista and MMBench, and even surpasses all closed-source models on MME, achieving a score of 2400. Besides, as shown in Fig. 3(b), by introducing the LLM-as-Judge for test-time scaling, VisionThink's answer outperforms the vanilla model's short direct answer. Moreover, we scale up the data size to 130K, and further demonstrate the effectiveness of LLM-as-Judge on General VQA Tasks. The results are shown in Appendix B.5.

### 4.3 Reinforcement Learning Enables VLM to Be More Efficient

**Comparison with the Reasoning Model.** To demonstrate the efficiency of our model, we first compare our VisionThink with QwenRL and QwenRL 1/4, both of which are reasoning models trained using the LLM-as-Judge strategy based on Qwen2.5-VL-7B Instruct. QwenRL and QwenRL 1/4 represent inference using the full-resolution image and the 1/4-resolution image, respectively. As shown in Fig. 4, we compare the inference time costs of the three models. Notably, the reported inference times reflect the actual time consumed during vLLM inference, which we believe best represents efficiency in real-world applications. The results show that on most benchmarks, our model's inference time is close to that of QwenRL 1/4, which uses 1/4 of the image tokens, and significantly better than the QwenRL model that processes all image tokens. Specifically, on the DocVQA benchmark, our VisionThink model is more than twice as fast as QwenRL. It also outperforms the baseline by approximately one-third in terms of inference time on benchmarks such as MME and POPE. It is worth noting that on strongly OCR-dependent benchmarks like ChartQA, our model consumes more time than the baseline QwenRL. This is because VisionThink identifies that most questions cannot be answered correctly at low resolution and thus autonomously requests high-resolution images. As a result, the total number of image tokens used by VisionThink exceeds that of the baseline, which we consider reasonable. However, such strongly OCR-dependent benchmarks are relatively rare, so the overall efficiency of VisionThink remains high.

**Comparison with the Previous Efficient VLM.** To further show the effectiveness of our Vision-Think, we compare it with the previous Efficient VLM method FastV and SparseVLM. Notably, all

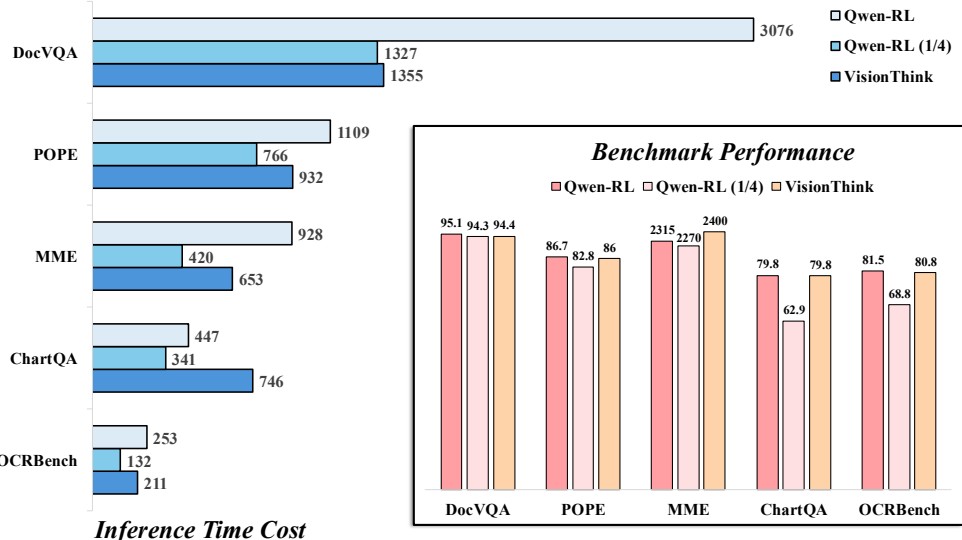

Figure 4: **Inference Time Cost (seconds) and Benchmark Performance Comparison for Reasoning Model**. Qwen-RL and Qwen-RL (1/4) represent leveraging the LLM-as-Judge on the Qwen2.5-VL-Instruct Model and inference on full resolution image and 1/4 resolution image, respectively.

these methods require computing attention scores to prune visual tokens, which makes them difficult to optimize with FlashAttention2 and may lead to increased memory usage. Furthermore, they are not directly compatible with the efficient inference framework vLLM. Therefore, to ensure a fair comparison, we evaluate model performance while keeping visual token consumption as consistent as possible. As shown in Table 2, our VisionThink outperforms previous methods on average across nine benchmarks. Furthermore, previous approaches require a predefined pruning ratio threshold, whereas our method can autonomously decide whether to reduce tokens based on the question and image content. As a result, on OCR-Related benchmarks such as ChartQA and OCR Bench, our method significantly surpasses FastV and SparseVLM by 9.0% and 8.3%, respectively.

## 4.4 Reinforcement Learning Enables VLM to Be Smarter

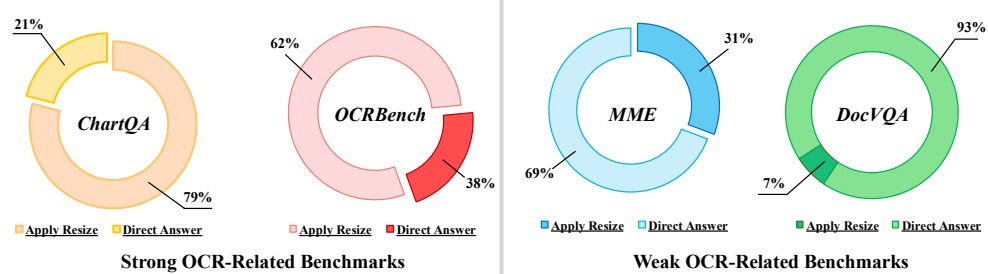

Figure 5: **VisionThink smartly determine the high-resolution image ratio.** Apply Resize indicates that the model autonomously requests to view the original high-resolution image, while Direct Answer indicates that the model is able to answer the question using only the 1/4-sized image.

In this section, we present the proportion of samples across different benchmarks for which our VisionThink gives direct answers versus those for which it requests high-resolution images. This illustrates the model's ability to smartly determine whether the information in the downsampled image is sufficient. As shown in Fig. 5, we observe that on benchmarks such as ChartQA and OCRBench, which require detailed visual understanding, our model shows a higher ratio of requests for high-resolution images. In contrast, for benchmarks like MME and DocVQA, at least 70% of

Table 2: **Comparison with Traditional Efficient VLM Methods.** Vanilla represents the Qwen2.5-VL-7B-Instrcut. The retained ratio of the baseline methods is a predefined hyperparameter, while for VisionThink, the ratio is determined autonomously by the model and reported as a statistical value. Note that *Down-Sample* refers to the model's performance when directly fed images with their resolution reduced by half. Additional baseline comparison results (VisionZip [80]) are shown in Table. 7

| Method | ChartQA[†] | OCRBench | DocVQA | MME | MMVet | RealWorldQA | POPE | MathVista | MathVerse | Avg. |
|--------|-----------|----------|--------|-----|-------|-------------|------|-----------|-----------|------|
| | test | test | val | test | test | test | test | testmini | testmini | |
| *Retain 100% Visual Tokens Across All Benchmarks* | | | | | | | | | | |
| Vanilla | 79.8 | 81.5 | 95.1 | 2316 | 61.6 | 68.6 | 86.7 | 68.2 | 46.3 | 100% |
| | 100% | 100% | 100% | 100% | 100% | 100% | 100% | 100% | 100% | |
| *Retain 25% Visual Tokens Across All Benchmarks* | | | | | | | | | | |
| Down-Sample | 62.9 | 68.8 | 94.3 | 2270 | 54.5 | 68.8 | 82.8 | 62.2 | 43.1 | 92.1% |
| | 78.8% | 84.4% | 99.1% | 98.0% | 88.5% | 100.3% | 95.5% | 91.2% | 93.1% | |
| *Retain 50% Visual Tokens Across All Benchmarks* | | | | | | | | | | |
| SparseVLM [100] | 73.2 | 75.6 | 66.8 | 2282 | 51.5 | 68.4 | 85.5 | 66.6 | 45.1 | **92.2%** |
| | 91.7% | 92.7% | 70.2% | 98.5% | 83.6% | 99.7% | 98.6% | 97.6% | 97.4% | |
| FastV [8] | 72.6 | 75.8 | 93.6 | 2308 | 52.8 | 68.8 | 84.7 | 63.7 | 45.0 | **95.8%** |
| | 91.0% | 93.0% | 98.4% | 99.6% | 85.7% | 100.3% | 97.7% | 93.4% | 97.2% | |
| *Retain 70% Visual Tokens Across All Benchmarks* | | | | | | | | | | |
| SparseVLM (ICML 2025) | 75.8 | 79.3 | 68.7 | 2276 | 53.7 | 68.5 | 85.4 | 66.3 | 45.1 | **93.6%** |
| | 94.9% | 97.3% | 72.2% | 98.3% | 87.2% | 99.8% | 98.5% | 97.2% | 97.4% | |
| FastV (ECCV 2024) | 77.2 | 82.2 | 94.4 | 2342 | 56.0 | 68.6 | 85.9 | 65.9 | 46.9 | **98.4%** |
| | 96.7% | 100.8% | 99.3% | 101.1% | 90.9% | 100% | 99.1% | 96.6% | 101.3% | |
| *Retain Approximately 51.3% Visual Tokens Across All Benchmarks* | | | | | | | | | | |
| VisionThink | 79.8 | 80.8 | 94.4 | 2400 | 68.5 | 67.1 | 86.0 | 67.5 | 48.0 | 101.4% |
| | 100% | 99.1% | 99.3% | 103.6% | 111.2% | 97.8% | 99.2% | 99.0% | 103.7% | |

the samples can be answered directly using low-resolution images at 1/4 of the original resolution. These results align with human intuition: most daily questions do not require high-resolution images, and only OCR-related tasks truly depend on them. Furthermore, to better demonstrate the 'smart' capabilities of VisionThink, we conduct case studies in Appendix D.

### 4.5 Relationship of the EfficientVLM methods and VisionThink.

**Key Differences.** Traditional EfficientVLM methods take a redundant image as input and attempt to remove the redundancy during inference. However, this process typically relies on fixed thresholds, which may yield acceptable performance on standard VQA tasks but result in poor performance on OCR-related or detail-sensitive scenarios, limiting their practical applicability. In contrast, VisionThink inputs reduced visual tokens and enables the model to autonomously determine whether a higher-resolution image is needed. Ideally, this approach avoids any performance degradation.

**Integration Potential.** Our proposed VisionThink essentially introduces a new paradigm for reading images, which can be integrated with existing Efficient VLMs. In this paper, to provide a straightforward validation of VisionThink, we chose to use image resizing perform token reduction. We believe that adopting more advanced token reduction techniques could further improve the model's direct answering accuracy, consequently, enhance its overall efficiency. Further discussions are shown in Appendix C.

## 5 Related Works

**Vision Language Model Reasoning.** With the advancement of LLM reasoning capabilities [17], many studies have aimed to improve the reasoning abilities of VLMs [98, 48, 43]. One common approach is using Chain-of-Thought (CoT) prompting to construct SFT datasets. However, the CoTs generated often lack natural human cognitive processes, limiting their effectiveness and generalization. Furthermore, inspired by DeepSeek-R1 [17], some studies have attempted to transfer this reasoning paradigm to vision tasks [87, 64, 20, 85]. However, most current approaches remain limited to the visual math and fail to generalize to general VQA tasks. In contrast, VisionThink successfully applies

effective reinforcement learning to general VQA by leveraging the LLM-as-Judge strategy. Due to space limitations, additional related work on efficient VLMs and LLM-based reasoning is presented in Appendix A.

# 6   Concluding Remarks

## 6.1   Summary

In this work, we introduce VisionThink, a novel paradigm for General VQA that enhances efficiency and performance. By initially processing a downsampled image and using reinforcement learning to selectively upscale to higher resolution when needed, VisionThink optimizes computational resources while preserving accuracy. Leveraging the LLM-as-Judge strategy and a tailored reward function, our approach outperforms prior state-of-the-art models across diverse VQA benchmarks, particularly in tasks requiring fine-grained details like OCR. We believe VisionThink demonstrates the potential of reinforcement learning in vision-language models and encourages the development of more effective and efficient AI systems.

## 6.2   Limitations and Future work

In this work, we focus on the setting of 2x resolution upscaling and at most two turns of conversations and yield promising results. However, it has not been extended to the setting of flexible resolution upscaling. Besides, incorporating more visual tools such as cropping would further bring benefits in both efficiency and performance. Furthermore, multi-turn (for example, more than 5 turns) image tool calls could gain more in solving complex visual problems.

Additionally, our paper utilizes image resizing to reduce the number of visual tokens. This simple method achieves a good balance between performance and efficiency via reinforcement learning. We hope this work inspires further research in the field of efficient reasoning vision language models, especially on making models smarter and more human-like. We will continue to explore the path toward building more general, powerful, and efficient vision-language models.

# 7   Acknowledgements

This work was supported in part by the Research Grants Council under the Areas of Excellence scheme grant AoE/E-601/22-R and the National Natural Science Foundation of China (No. 62422606, 62201484).

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

# Contents

# A   Related Works

## A.1   Efficient Vision Language Models

Large Language Models (LLMs) have demonstrated remarkable progress in language understanding and generation [1, 66, 4, 51, 13, 29]. Building on their success, VLMs have rapidly advanced by integrating visual information into LLM architectures [33, 34, 31, 65, 67, 35, 102, 26, 82]. Prominent models such as LLaVA [33] utilize visual encoders followed by the projection layers to convert images into token sequences compatible with LLMs. However, as the performance of vision-language models continues to improve, the number of visual tokens grows rapidly, leading to increased computational costs. This trend limits the practical deployment of such models in scenarios like edge computing, autonomous driving, medical analysis, and robotics [24, 36, 54, 81, 83, 89, 55, 61, 52, 99, 53, 68, 84]. Therefore, it is imperative to avoid the excessive use of visual tokens.

Recently, some studies [8, 100, 77, 71, 59, 18, 80, 101] have also recognized the redundancy in visual tokens and proposed various methods to address it. Most of these works input images containing redundancy and use the attention scores assigned by the model to prune or merge tokens for token reduction. Furthermore, they typically apply a fixed threshold to reduce the same proportion of redundant tokens across all data samples. Although these methods maintain good performance on general VQA tasks, they perform poorly on OCR-related benchmarks. In contrast to previous works, our proposed VisionThink initially inputs reduced tokens and allows the model to autonomously

determine whether token reduction is sufficient or if a high-resolution image is required. Through this approach, our method achieves efficiency while maintaining strong performance on OCR-related benchmarks. Additionally, VisionThink is not a specific token-level reduction strategy but represents a new paradigm that can be integrated with existing EfficientVLM methods.

## A.2 Large Language Model Reasoning

Recent advances in large language models (LLMs) [70, 44, 49, 63, 17, 73, 62, 79] have significantly improved their reasoning capabilities through methods that simulate human-like stepwise thinking. One foundational technique, Chain-of-Thought (CoT) prompting [69], encourages models to decompose complex tasks into intermediate steps, enhancing performance on a variety of reasoning benchmarks. Furthermore, researchers have explored more structured and dynamic reasoning paradigms, such as Tree-of-Thought and Graph-of-Thought [88, 7], which organize reasoning as branching or interconnected processes. Complementary approaches like Program-of-Thought (PoT) [10] further improve reasoning fidelity by integrating external computational tools to verify or simplify logic steps.

Besides, recent work has also shifted attention from model architecture design and train-time scaling to test-time scaling [60], such as Monte Carlo Tree Search (MCTS) [76], stepwise preference optimization [27], and reinforcement learning [44] are used to refine outputs during inference. Models such as DeepSeek-R1 [17], OpenAI-O1 [49] demonstrate the effectiveness of combining large-scale RL with reward functions that prioritize both correctness and reasoning quality. Although LLMs have shown remarkable progress in structured reasoning, extending these abilities to Vision Language Models remains an open challenge.

## A.3 Vision Language Model Reasoning

With the advancement of LLM reasoning capabilities [17], many studies have aimed to improve the reasoning abilities of VLMs [98, 48, 43]. One common approach is using Chain-of-Thought (CoT) prompting to construct SFT datasets. However, the CoTs generated often lack natural human cognitive processes, limiting their effectiveness and generalization. Furthermore, inspired by DeepSeek-R1 [17], several studies have attempted to transfer this reasoning paradigm to vision tasks [87, 64, 20, 85, 38, 57, 47, 41, 72, 40, 39]. Most of these efforts, by collecting CoT data to perform a cold start and then training the model using a reinforcement learning strategy such as GRPO. While this approach achieves performance improvements on specific tasks, it significantly degrades the model's general performance. Moreover, current methods remain limited to visual math or segmentation tasks, failing to generalize to broader general VQA tasks. In this paper, we propose VisionThink, which effectively applies reinforcement learning to general VQA tasks by leveraging the LLM-as-Judge strategy.

# B  Additional Experiments

## B.1  Prompt Details

### B.1.1  LLM-as-Judge Prompt Design

In this section, we detail the prompt design for our LLM-as-Judge strategy. As shown in Table 3, the placeholders Ground Truth and Prediction are dynamically replaced with the corresponding question, ground truth answer, and model prediction during evaluation. Specifically, the judgment process is conducted entirely in text. Our findings indicate that, compared to VLMs, current LLMs achieve higher judgment accuracy and exhibit fewer hallucinations. Moreover, by eliminating the need for visual token inputs, it significantly reduce the overall evaluation cost.

Furthermore, we require the LLM to return a discrete value, with 1 indicating a correct prediction and 0 indicating an incorrect one, rather than a continuous score representing the degree of correctness. This binary format further reduces the likelihood of misjudgment. In a user study of 1,000 cases, no misclassifications were observed.

Table 3: **Judgment Prompt Template.** Question, Ground Truth and Prediction are dynamically replaced with the specific question, ground truth and model prediction during evaluation.

*SYSTEM PROMPT:*
You are an intelligent chatbot designed for evaluating the correctness of generative outputs for question-answer pairs.
Your task is to compare the predicted answer with the correct answer and determine if they match meaningfully. Here's how you can accomplish the task:
INSTRUCTIONS:
- Focus on the meaningful match between the predicted answer and the correct answer.
- Consider synonyms or paraphrases as valid matches.
- Evaluate the correctness of the prediction compared to the answer.

*USER PROMPT:*
I will give you a question related to an image and the following text as inputs:
1. **Question Related to the Image**: Question
2. **Ground Truth Answer**: Ground Truth
3. **Model Predicted Answer**: Prediction
Your task is to evaluate the model's predicted answer against the ground truth answer, based on the context provided by the question related to the image. Consider the following criteria for evaluation:
- **Relevance**: Does the predicted answer directly address the question posed, considering the information provided by the given question?
- **Accuracy**: Compare the predicted answer to the ground truth answer. You need to evaluate from the following two perspectives:
(1) If the ground truth answer is open-ended, consider whether the prediction accurately reflects the information given in the ground truth without introducing factual inaccuracies. If it does, the prediction should be considered correct.
(2) If the ground truth answer is a definitive answer, strictly compare the model's prediction to the actual answer. Pay attention to unit conversions such as length and angle, etc. As long as the results are consistent, the model's prediction should be deemed correct.
**Output Format**:
Your response should include an integer score indicating the correctness of the prediction: 1 for correct and 0 for incorrect. Note that 1 means the model's prediction strictly aligns with the ground truth, while 0 means it does not.
The format should be Score: 0 or 1

### B.1.2  VisionThink Image Resize Prompt

As shown in Table 4, we present the detailed system and user prompts used in our proposed Vision-Think. Specifically, we integrate image resizing as a tool-call function. Following the Qwen2.5-VL cookbook [6], we employ an Agent Prompt that enables the model to output special tokens to trigger image resizing. This prompt design allows the model to exhibit distinct behaviors such as requesting image resizing or directly answering the question. These behaviors introduce differentiable gradients, which make it feasible to apply the GRPO algorithm. Furthermore, we analyze the impact of different prompts in Sec. C.3.

### B.2  Details of the Format Reward

The format reward has a total score of 0.5, which is awarded only when all formatting requirements are fully satisfied. Specifically, as shown in the VisionThink Prompt (Table. 4), the first requirement is that the model's output must include both the the the <answer></answer> and <think></think> tags, which denote the final answer and the reasoning process, respectively. The second requirement states that for responses involving an image resize operation, the model must output a correctly formatted <tool_call></tool_call> tag containing a valid JSON content.

Table 4: **VisionThink Image Resize Prompt Template.** Question will be replaced with the specific question during training and inference.

---

*SYSTEM PROMPT:*
You are a helpful assistant.
# Tools
You may call the function tool shown below to assist with the user query.
You are provided with the function signature within <tools></tools> XML tags:
<tools>
{
    "type": "function",
    "function":{
      "name_for_human": "resize_image",
      "name": "resize_image",
      "description": "Resize the image resolution.",
        "parameters": {
        "properties": {
          "action": {
            "description": "The action to perform. The available actions are:
                **resize**: Double the resolution of the current image. You should only use this tool if you are unable to obtain the critical information needed to answer the question from the current resolution.",
            "enum": ["resize"],
            "type": "string"
          }
        }
      "required": ["action"],
      "type": "object",
    },
    "args_format": "Format the arguments as a JSON object."
    }
}
</tools>
For each function call, return a json object with the function name and the corresponding argument within <tool_call></tool_call> XML tags:
<tool_call> {"name": <function-name>, "arguments": <args-json-object>} </tool_call>

---

*USER PROMPT:*
Answer the question based on the image provided. You must conduct reasoning within `<think>` and `</think>` first in each of your reasoning steps. You may call ONE function tool per step to help you better solve the problem. Place the function tool within `<tool_call>` and `</tool_call>` at the end of each step to perform a function call. You should continue your reasoning process based on the content returned by the function tool. Once you confirm your final answer, place the final answer inside `<answer>` and `</answer>`. For mathematical or multiple-choice problem, wrap the answer value or choice with `\boxed{}`. Here is the image and question: Question.

---

## B.3 Implementation Details

**Training Details.** In this paper, we conduct experiments using Qwen2.5-VL-7B-Instruct [5] as the base model, trained with the veRL framework [58]. We use a total batch size of 512 with mixed-precision (FP16) training. The mini-batch size is set to 32, and the KL divergence coefficient is 0.001. The policy model is optimized using an initial learning rate of $1 \times 10^{-6}$. For each prompt, we generate 16 candidate responses using a temperature of 1.0, and apply duplicate and empty response filtering, similar to DAPO [90].

**Inference Details.** In this paper, we use the lmms-eval [94] to evaluate the model's performance. Besides, in order to save the GPU memory and improve the inference speed, we utilize the vLLM[25] framework and set the temperature to zero for inference.

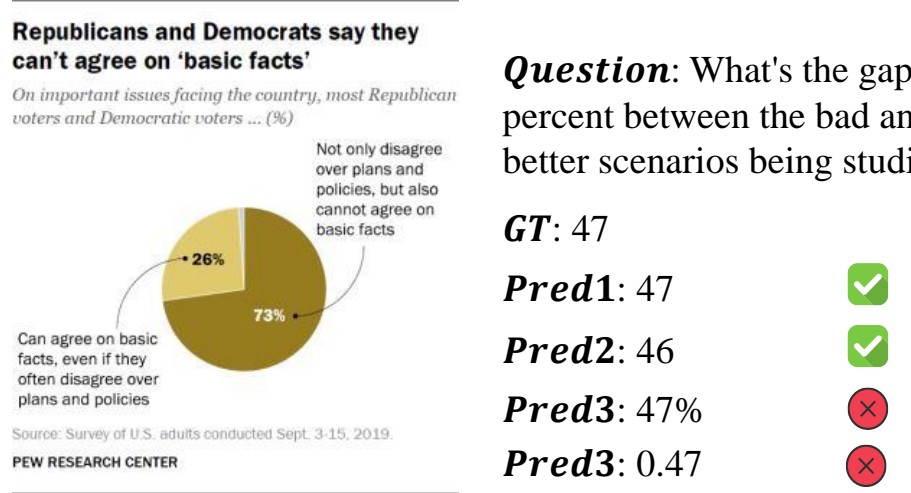

Figure 6: An example illustrating the original evaluation method used in ChartQA.

### B.4 Benchmark Datasets and Evaluation Metrics

We conduct experiments on these widely used visual understanding benchmarks.

**ChartQA.** ChartQA [45] is a benchmark designed to evaluate how well multimodal models answer questions about charts, emphasizing both visual understanding and logical reasoning. It includes various chart types, such as bar charts and line graphs, with a mix of human-written and automatically generated questions to assess complex reasoning abilities. Notably, ChartQA is a strongly OCR-dependent benchmark that requires fine-grained visual understanding, as models must extract textual information from charts and reason over it.

```python
def _to_float(text: str):
    try:
        if text.endswith("%"):
            return float(text.rstrip("%")) / 100.0
        else:
            return float(text)
    except ValueError:
        return None

prediction_float = _to_float(prediction)
target_float = _to_float(target)

if prediction_float is not None and target_float is not None:
    relative_change = abs(prediction_float - target_float) / abs(
        target_float)
    return relative_change <= max_relative_change   # 0.05
else:
    return prediction.lower() == target.lower()
```

Listing 1: Core evaluation code from the original ChartQA assessment method..

Furthermore, we observe that the evaluation process of ChartQA in lmms-eval [94] relies on a float-value comparison method, which presents several limitations in practical evaluation scenarios. The corresponding implementation is shown in Listing 1, and an illustrative example is provided in Fig. 6 for further analysis.

As shown in Fig. 6, for the question "What's the gap percent between the bad and the better scenarios being studied?", the intuitive answer derived from the image is 47%. And the `_to_float()` function (Line 1 in Listing 1) converts both `0.47` and `47%` to `0.47`, while converting the ground truth value `47` to `47.0`. Hence, the comparison at Line 14 treats both `0.47` and `47%` as incorrect predictions, leading

to an erroneous evaluation result. Moreover, when the model incorrectly predicts 46, the current evaluation method still considers it correct, as the relative error compared to the ground truth 47 is:

$$\frac{|46 - 47|}{47} = 0.02 < 0.05.$$

which is also a wrong judgment result of the evaluation.

Based on this observation, all ChartQA evaluations in this paper are conducted using a combination of GPT-4o-Judge and human verification, denoted as ChartQA$^{\dagger}$.

**MME.** The MME benchmark [15] assesses multimodal models on 14 subtasks that reflect both perceptual processing and cognitive reasoning abilities. By utilizing carefully crafted instruction-response pairs, MME aims to minimize the risk of training data contamination, ensuring a fair and rigorous evaluation process.

**OCRBench.** OCRBench [37] is a comprehensive benchmark for evaluating the OCR capabilities of vision language models. It covers five key tasks: text recognition, scene text-centric VQA, document-oriented VQA, key information extraction, and handwritten mathematical expression recognition. With 29 datasets and 10,000 human-verified QA pairs across 31 scenarios. Its scenarios span street scenes, receipts, and formulas, testing models on multilingual, handwritten, non-semantic, and mathematical text.

**DocVQA.** DocVQA [46] is a dataset for VQA on document images, comprising 50,000 questions defined on over 12,000 document images. It covers various document types, including forms, receipts, and scientific papers, testing models' ability to understand and reason about document content, such as textual information, tables, and visual elements.

**RealWorldQA.** RealWorldQA [74] is a benchmark designed to evaluate the real-world spatial understanding capabilities of VLMs. It consists of over 700 images, each accompanied by a question and a verifiable answer, drawn from real-world scenarios, including those captured from vehicles. The benchmark assesses how well models comprehend physical environments and spatial relationships, which are crucial for applications in navigation, robotics, and general AI assistance.

**MMVet.** MMVet [92] introduces a structured framework to assess six foundational vision-language skills: recognition, OCR, knowledge, language generation, spatial awareness, and math. These capabilities are combined in 16 evaluation configurations to test how well multimodal systems can integrate them for solving complex tasks, offering a detailed and quantitative performance analysis.

**POPE.** POPE [32] is designed to measure object hallucination in vision-language models using binary-choice questions that verify whether specific objects are present in given images. It employs metrics such as Accuracy, Recall, Precision, and F1 Score across three distinct sampling strategies, delivering a robust and fine-grained evaluation of hallucination tendencies. In our paper, the result of POPE is F1-score.

**MMMU.** MMMU [93] serves as a benchmark for assessing multimodal models on intricate, college-level tasks that demand both extensive knowledge and reasoning capabilities. It comprises 11.5K carefully selected questions sourced from exams, quizzes, and academic textbooks, spanning six broad fields: Art & Design, Business, Science, Health & Medicine, Humanities & Social Sciences, and Technology & Engineering. These questions encompass 30 academic subjects and 183 specialized areas, incorporating a wide variety of visual formats such as diagrams, graphs, and chemical formulas. MMMU is designed to push models toward expert-level performance by testing their ability to understand and reason across disciplines and modalities.

**MathVista.** MathVista [42] is a benchmark for evaluating the mathematical reasoning capabilities of foundation models within visual contexts. It includes 6,141 examples, derived from 28 existing multimodal datasets involving mathematics and three newly created datasets: IQTest, FunctionQA, and PaperQA. These tasks require fine-grained visual understanding and compositional reasoning, often involving the interpretation of graphs, equations, and other mathematical visuals. MathVista aims to systematically study the ability of VLMs to solve mathematical problems presented in visual formats, highlighting the need for models that can seamlessly integrate visual perception with mathematical reasoning.

**MathVerse.** MathVerse [96] is a benchmark for rigorously evaluating the capabilities of VLMs in interpreting and reasoning with visual information in mathematical problems. MathVerse consists of 2,612 high-quality, multi-subject math problems with diagrams, each transformed into six distinct versions with varying degrees of information content in multi-modality, resulting in 15,000 test samples.

### B.5  Scaling-up Reinforcement Learning on General VQA Tasks

Due to the diversity and complexity inherent in general VQA tasks, traditional rule-based reinforcement learning algorithms are not directly applicable. To overcome this limitation, we introduce an LLM-as-Judge strategy, which enables our model to be trained via reinforcement learning on the General VQA task. To further demonstrate the effectiveness of our method, we scale up the dataset size to 130K to validate its effectiveness.

**Dataset.** Since the LLM-as-Judge approach is flexible, one advantage is that most of the SFT data can be utilized. Therefore, we only filter out subjective open-ended questions whose answers are not unique and can be correctly addressed from different perspectives, such as image descriptions, essay writing, and similar tasks. Based on this, we ultimately filtered 130K QA pairs to train the VLM via reinforcement learning, without requiring any cold-start phase. All the data will be open-sourced.

**Prompt.** To verify the effectiveness of the LLM-as-Judge strategy on general VQA tasks, we conduct experiments with minimal modifications to both the system and user prompts. The detailed prompts are provided in Table 5.

**Reward.** Since the entire training process in this setting does not involve any decision-making regarding the need for high-resolution images, the total reward function in reinforcement learning is designed to focus solely on answer quality and response formatting. Specifically, the reward comprises two components: The first component is an accuracy reward, evaluated by the LLM-as-Judge. This component assesses the correctness of the model's answer against the ground truth, with a maximum of 1 point awarded for a fully correct response. The second component is a formatting reward, worth 0.5 points. This is granted when the model correctly wraps its response using both the <answer></answer> and <think></think> tags. These tags are critical for maintaining consistent output formatting and enabling downstream interpretability.

Table 5: **Prompt Template for VisionThink♠.** VisionThink♠ refers to a model trained on general VQA tasks using full image resolution and the LLM-as-Judge strategy. The Question placeholder is replaced with the actual question during training and inference.

| |
|---|
| *SYSTEM PROMPT:* |
| You FIRST think about the reasoning process as an internal monologue and then provide the final answer. |
| The reasoning process MUST BE enclosed within `<think> </think>` tags. The final answer MUST BE put within `<answer> </answer>` tags. For mathematical or multiple-choice problems, wrap the answer value or choice with `\boxed{}`. |
| *USER PROMPT:* |
| Question. |

**Experimental Results.** As shown in Table 6, we compare our model against state-of-the-art open-source and closed-source vision-language models across several general VQA benchmarks. In this evaluation, VisionThink ♠ denotes our model variant trained using the proposed LLM-as-Judge strategy with the above reward function and 130K QA pairs.

The experimental results demonstrate that our method outperforms the baseline model, Qwen2.5VL-7B-Instruct, across multiple benchmarks. The improvement is particularly notable on the MMVet, where our model achieves a significant performance gain of 7.9% over the baseline. This highlights the model's superior capability in handling general VQA tasks. Furthermore, on the recently popular benchmark MathVista [42], which is designed to assess both mathematical reasoning and general visual question answering reasoning abilities, our model achieves a score of 71.2. This result not only surpasses all existing open-source models but also outperforms several closed-source models.

Table 6: **Effectiveness of LLM-as-Judge Accuracy Reward Design.** VisionThink♠ is a model we developed by training with the LLM-as-Judge on 130K filtered General VQA datasets and leverages Qwen2.5-VL-7B-Instruct as the base model. Qwen2.5-VL-7B* reports the results evaluate by lmms-eval[94].

| Method | MMMU | MMMU-Pro | MMBench | RealWorldQA | POPE | MME | MathVista | MathVerse | MMVet |
|---|---|---|---|---|---|---|---|---|---|
| | val | test | en_test | test | test | test | testmini | testmini | test |
| *Closed-Source Model* | | | | | | | | | |
| GPT-4o [50] | 69.1 | 54.0 | 83.4 | 58.6 | 85.6 | 2329 | 63.8 | 50.2 | 69.1 |
| Claude-3.5 Sonnet [2] | 68.3 | 55.0 | 82.6 | 59.9 | - | 1920 | 67.7 | 41.2 | 70.1 |
| Gemini-1.5-Pro [62] | 62.2 | 49.4 | 73.9 | 70.4 | 88.2 | - | 63.9 | - | 64.0 |
| *Open-Source General Model* | | | | | | | | | |
| Cambrain-1-8B [65] | 42.7 | - | 75.9 | 60.0 | 86.4 | 1803 | 49.0 | - | - |
| InternVL2-8B [12] | 49.3 | 32.5 | 81.7 | 64.4 | 84.2 | 2210 | 58.3 | - | 60.0 |
| LLaVA-OneVision-7B [28] | 48.8 | - | - | 66.3 | 88.4 | 1998 | 63.2 | - | 57.5 |
| MiniCPM-Llama-V-2.5-8B [89] | 45.8 | 19.6 | 77.2 | 63.0 | 86.7 | 2025 | 54.3 | - | - |
| MiniCPM-V-2.6-8B [89] | 49.8 | 27.2 | 78.0 | 65.0 | 83.2 | 2348 | 60.6 | - | - |
| IXC-2.5 [95] | 42.9 | - | 82.2 | 67.8 | - | 2229 | 63.8 | - | 51.7 |
| InternVL2.5-8B [11] | 56.0 | 38.2 | 84.6 | 70.1 | 90.6 | 2344 | 64.4 | 39.5 | 62.8 |
| *Reasoning Model* | | | | | | | | | |
| LLaVA-CoT-11B [78] | - | - | 75.0 | - | - | - | 54.8 | - | 60.3 |
| LLaVA-Reasoner-8B [97] | - | - | - | - | - | - | 50.6 | - | - |
| Insight-V-8B [14] | 50.2 | 24.9 | 82.3 | - | - | 2312 | 59.9 | - | - |
| Mulberry-7B [86] | 55.0 | - | - | - | - | 2396 | 63.1 | - | - |
| Vision-R1-LlamaV-CI-11B [19] | - | - | - | - | - | 2190 | 62.7 | 27.1 | - |
| *VisionThink* | | | | | | | | | |
| Qwen2.5-VL-7B* [5] | 50.3 | 37.7 | 82.6 | 68.6 | 86.7 | 2316 | 68.2 | 46.3 | 61.6 |
| VisionThink ♠ | **52.7** | **41.1** | **83.4** | 66.5 | **88.6** | 2314 | **71.2** | **48.3** | **69.5** |

These findings provide strong empirical evidence for the effectiveness and generalizability of our LLM-as-Judge strategy in enhancing the reasoning capabilities of VLMs across general VQA tasks.

## B.6 Comparison with Previous Efficient VLM

To further demonstrate the effectiveness of our proposed VisionThink, we conduct a comparative analysis against an additional efficient Vision-Language Model (VLM), VisionZip [80]. While previous methods such as FastV [8] and SparseVLM [100] perform token reduction within the language model component based on attention scores, VisionZip applies reduction directly within the vision encoder using a similar attention-based mechanism.

As shown in Table 7, although previous efficient VLM methods achieve competitive performance on general VQA benchmarks, their accuracy drops significantly on OCR-related tasks. This degradation is particularly evident even when a substantial portion of the visual token is retained (70%), as demonstrated on the ChartQA dataset.

In contrast, our proposed model, VisionThink, can smartly decide whether to request the original high-resolution image based on the complexity and demands of each sample. This adaptive strategy enables the model to maintain high accuracy on general VQA tasks while substantially improving performance on benchmarks requiring detailed textual recognition. Through this capability, VisionThink demonstrates stronger fine-grained visual understanding and addresses a key limitation of previous efficient VLMs—namely, their poor performance on OCR-related tasks, which has constrained their applicability in real-world scenarios.

Notably, VisionZip‡ refers to the variant fine-tuned on the 130K dataset [80]. However, compared to the training-free version of VisionZip, this fine-tuned model does not show any performance improvement. We attribute this to the limited coverage and diversity of the fine-tuning dataset,

Table 7: **Comparison with Previous Efficient VLM Methods.** Vanilla represents the Qwen2.5-VL-7B-Instrcut. The retained ratio of the baseline methods is a predefined hyperparameter, while for VisionThink, the ratio is determined autonomously by the model and reported as a statistical value. Note that *Down-Sample* refers to the model's performance when directly fed images with their resolution reduced by half. VisionZip‡ represents using the 130K data to finetuning the model.

| Method | ChartQA† | OCRBench | DocVQA | MME | MMVet | RealWorldQA | POPE | MathVista | MathVerse | Avg. |
|---|---|---|---|---|---|---|---|---|---|---|
| | test | test | val | test | test | test | test | testmini | testmini | |
| *Retain 100% Visual Tokens Across All Benchmarks* | | | | | | | | | | |
| Vanilla | 79.8 | 81.5 | 95.1 | 2316 | 61.6 | 68.6 | 86.7 | 68.2 | 46.3 | 100% |
| | 100% | 100% | 100% | 100% | 100% | 100% | 100% | 100% | 100% | |
| *Retain 25% Visual Tokens Across All Benchmarks* | | | | | | | | | | |
| Down-Sample | 62.9 | 68.8 | 94.3 | 2270 | 54.5 | 68.8 | 82.8 | 62.2 | 43.1 | 92.1% |
| | 78.8% | 84.4% | 99.1% | 98.0% | 88.5% | 100.3% | 95.5% | 91.2% | 93.1% | |
| *Retain 50% Visual Tokens Across All Benchmarks* | | | | | | | | | | |
| FastV (ECCV 2024) | 72.6 | 75.8 | 93.6 | 2308 | 52.8 | 68.8 | 84.7 | 63.7 | 45.0 | **95.8%** |
| | 91.0% | 93.0% | 98.4% | 99.6% | 85.7% | 100.3% | 97.7% | 93.4% | 97.2% | |
| SparseVLM (ICML 2025) | 73.2 | 75.6 | 66.8 | 2282 | 51.5 | 68.4 | 85.5 | 66.6 | 45.1 | **92.2%** |
| | 91.7% | 92.7% | 70.2% | 98.5% | 83.6% | 99.7% | 98.6% | 97.6% | 97.4% | |
| VisionZip (CVPR 2025) | 73.4 | 70.5 | 93.8 | 2209 | 57.0 | 68.6 | 86.3 | 63.7 | 45.1 | **95.0%** |
| | 92.0% | 86.5% | 98.6% | 95.4% | 92.5% | 100% | 99.5% | 93.4% | 97.4% | |
| VisionZip‡ (CVPR 2025) | 77.3 | 77.9 | 93.8 | 2244 | 50.1 | 69.2 | 91.2 | 63.1 | 39.4 | **95.0%** |
| | 96.9% | 95.6% | 98.6% | 96.9% | 81.3% | 100.9% | 107.5% | 92.5% | 85.1% | |
| *Retain 70% Visual Tokens Across All Benchmarks* | | | | | | | | | | |
| FastV (ECCV 2024) | 77.2 | 82.2 | 94.4 | 2342 | 56.0 | 68.6 | 85.9 | 65.9 | 46.9 | **98.4%** |
| | 96.7% | 100.8% | 99.3% | 101.1% | 90.9% | 100.0% | 99.1% | 96.6% | 101.3% | |
| SparseVLM (ICML 2025) | 75.8 | 79.3 | 68.7 | 2276 | 53.7 | 68.5 | 85.4 | 66.3 | 45.1 | **93.6%** |
| | 94.9% | 97.3% | 72.2% | 98.3% | 87.2% | 99.8% | 98.5% | 97.2% | 97.4% | |
| VisionZip (CVPR 2025) | 76.8 | 80.9 | 94.5 | 2334 | 60.0 | 68.2 | 86.4 | 68.9 | 45.8 | **99.1%** |
| | 96.2% | 99.3% | 99.4% | 100.8% | 97.4% | 99.4% | 99.7% | 101.0% | 98.9% | |
| VisionZip‡ (CVPR 2025) | 78.2 | 81.3 | 94.1 | 2230 | 52.5 | 68.6 | 92.5 | 64.8 | 41.8 | **96.7%** |
| | 98.0% | 99.8% | 98.9% | 96.3% | 85.3% | 100% | 106.7% | 95.0% | 90.3% | |
| *Retain Approximately 51.3% Visual Tokens Across All Benchmarks* | | | | | | | | | | |
| VisionThink | 79.8 | 80.8 | 94.4 | 2400 | 68.5 | 67.1 | 86.0 | 67.5 | 48.0 | **101.4%** |
| | 100.0% | 99.1% | 99.3% | 103.6% | 111.2% | 97.8% | 99.2% | 99.0% | 103.7% | |

which falls short of the supervised fine-tuning data used by the official Qwen team. This observation indirectly suggests that, compared to supervised fine-tuning, reinforcement learning provides better generalization, which we further discuss in Sec. C.1.

### B.7 Additional Discussion Experiments

**Stronger perception tasks.** We adopt the widely used CV-Bench, introduced in Cambrian-1 [65], and follow its official setting and prompt for the counting task. Besides, we do not introduce any additional data for task-specific training. All models used are same to those in the main paper.

As shown in Table 8a, both VisionThink and VisionThink‡ outperform the base model (Qwen2.5VL-7B) on the counting benchmark, demonstrating that our approach retains strong performance even on stronger visual perception tasks.

**Compared to the keyword-based method.** Our VisionThink can automatically detect when the visual information is insufficient and decide when to resize the image accordingly. This naturally raises an interesting question: could we leverage keyword-based detection to determine when a large-sized image is needed to provide more information, and when a smaller image would be sufficient?

Table 8: **Additional Discussion Experiments.**

(a) Results on stronger perception tasks.

| Task | Qwen2.5VL-7B | VisionThink | VisionThink‡ |
|---|---|---|---|
| Counting | 63.1% | 65.7% | 67.4% |

(b) Comparison with Keyword-Based method.

| Task | Qwen-RL | VisionThink | Keyword-Based |
|---|---|---|---|
| ChartQA | 79.8% | 79.8% | 67.6% |

Table 9: VisionThink results on MiMO-VL.

| MiMo-VL | ChartQA | OCRBench | MME |
|---|---|---|---|
| Vanilla | 91.3 | 86.6 | 2330 |
| Down-Sample | 69.8 | 73.1 | 2300 |
| **VisionThink (MiMo)** | 88.7 | 86.5 | 2326 |

Hence, to further assess the value of VisionThink, we compare it against a keyword-based resolution selection approach. First, we use GPT-4o to generate 100 single-word fine-grained/OCR-related keywords (e.g., counting, value, locate) and 100 short phrases (e.g., how many, fine detail). Then, the system defaults to Qwen-RL (1/4) for efficient inference. When a keyword is detected in the question, it switches to full-resolution inference via Qwen-RL. This simulates a keyword-triggered token selection policy.

As shown in Table 8b, due to the diversity in question phrasing, keyword-based strategies generalize poorly and result in suboptimal performance. Moreover, in real-world deployment, VisionThink only requires deploying a single model, while keyword-based approaches require maintaining two separate models, leading to increased resource consumption.

**Adding VisionThink to additional VLMs.** Recently, the MiMO team proposed MiMO-VL-SFT [75], which achieves strong performance across several benchmarks. To further demonstrate the generalization ability of our proposed VisionThink, we integrate it into MiMO-VL. As shown in Table 9, our method also achieves strong performance on MiMO-VL, highlighting the broad applicability and generalization capability of VisionThink.

**VisionThink trained with rule-based reward on easily verifiable tasks.** To explore whether our proposed method can be further optimized on easily verifiable tasks using rule-based approaches, we conduct additional investigations.

*Datasets.* We filtered structured QA samples from our training set where answers can be validated using huggingface/math-verify or string match. These samples are primarily from OCR-related datasets, where answer verification is reliable.

*Reward.* Instead of relying on the LLM-as-Judge for binary rewards, we use rule-based verification huggingface/math-verify and string match to provide the reward.

*Results.* As shown in Table 10, the first two models (Qwen-RL and VisionThink) are same as the main paper that trained with LLM-as-Judge. The final column shows VisionThink (Rule-Based) trained via rule-based reward only. It maintains strong accuracy and efficiency on verifiable tasks like ChartQA and DocVQA.

*Discussion.* We used the LLM-as-Judge in the main paper to handle general QA scenarios, where reliable rule-based supervision is difficult to define. However, as this experiment shows, for easily verifiable tasks such as OCR-related QA, VisionThink can be effectively trained with rule-based reinforcement learning alone.

**Results on more fine-grained benchmarks.** In the main paper, ChartQA, OCRBench, and MathVista are fine-grained benchmarks that typically require the model to resize images to obtain more detailed information. In contrast, benchmarks such as DocVQA, MME, RealWorldQA, and POPE represent more general scenarios where image resizing is unnecessary, as low-resolution images are sufficient for the model to complete the tasks.

Table 10: VisionThink trained with rule-based reward.

| Task | Metric | Qwen-RL | VisionThink | VisionThink (Rule-Based) |
|------|--------|---------|-------------|--------------------------|
| ChartQA | Accuracy | 79.8 | 79.8 | 80.8 |
|  | Time (s) | 447 | 746 | 778 |
| DocVQA | Accuracy | 95.1 | 94.4 | 94.7 |
|  | Time (s) | 3076 | 1355 | 1824 |
| OCRBench | Accuracy | 81.5 | 80.8 | 79.8 |
|  | Time (s) | 253 | 211 | 183 |

Table 11: Results on more fine-grained benchmarks.

| Model | V$^*$ Bench | MME-RealWorld-Lite | HR-Bench-4K | HR-Bench-8K | TreeBench | Avg. |
|-------|-------------|--------------------|-------------|-------------|-----------|------|
| *Resolution (W×H)* | 2246×1583 | 2076×1434 | 4023×3503 | 5727×4430 | 2152×1615 | |
| *Token Usage* | 43% | 110% | 58.0% | 51.0% | 115% | |
| Vanilla | 72.3 | 45.1 | 71.4 | 67.6 | 39.5 | 100% |
| Down-Sample | 69.0 | 39.4 | 69.4 | 66.0 | 37.3 | 94.4% |
| **VisionThink** | 72.3 | 48.4 | 70.2 | 67.3 | 42.5 | **102.6%** |

In this section, we explore the model's performance on more fine-grained benchmarks, including V* Bench, MME-RealWorld-Lite, HR-Bench-4K, HR-Bench-8K, and TreeBench, all of which contain high-resolution images with relatively large image sizes.

As shown in Table 11, on HR-Bench, where image resolutions are extremely high, we observe that even the downsampled versions maintain sufficient visual clarity. Consequently, performance does not degrade significantly when the resolution is reduced. On MME-RealWorld-Lite and TreeBench, our VisionThink not only substantially outperforms the Down-Sample baseline (which uses the same input resolution), but even slightly surpasses Vanilla, which performs inference on 2× higher-resolution inputs. We hypothesize that this phenomenon occurs because, in cases where upscaling is triggered, the model effectively views the same image at two different resolutions, thereby gaining a dual-perspective understanding. This process may serve as a form of implicit data augmentation, leading to improved performance. Overall, this is an exciting finding—it suggests that VisionThink not only enhances efficiency in general scenarios but also has the potential to improve performance on fine-grained tasks.

## C  Further Discussions

### C.1  Why Use RL Instead of SFT?

In this paper, we train a smart and efficient vision-language model via reinforcement learning. A natural question arises: why use reinforcement learning instead of supervised fine-tuning to achieve this goal?

To answer this question, we conduct a comparative SFT experiment. Firstly, we construct the SFT training set. Specifically, compared to RL, which can directly utilize QA pairs and autonomously learn both the reasoning process and whether a high-resolution image is needed, SFT requires manually crafting both the reasoning steps and dialogue that involves high-resolution image requests. To overcome this limitation, we use GPT-4o to simulate both the high-resolution image requests and the corresponding reasoning process, enabling the SFT training data to closely approximate the behavior of the RL-trained model. Finally, we convert the original RL training data into a format compatible with SFT, maintaining a 1:1 ratio between high-resolution requests and direct answers.

As shown in Table 12, we compare the proportion of high-resolution image requests made by the SFT and RL models across evaluation benchmarks. Compared to the RL-trained model, which can smartly

Table 12: Comparison of image resize call ratios for RL and SFT trained models over multiple evaluation benchmarks.

| Method | ChartQA$^\dagger$ | OCRBench | DocVQA | MME | RealWorldQA | POPE |
|---|---|---|---|---|---|---|
| RL | 79.1% | 62.3% | 6.5% | 30.7% | 29.9% | 9.5% |
| SFT | 95.1% | 64.0% | 14.1% | 39.0% | 62.1% | 37.8% |

Table 13: **Performance comparison of the cold start model and no cold start model.** The without cold start model represents our VisionThink.

| Method | Type | ChartQA$^\dagger$ | OCRBench | DocVQA | MME | MMVet | RealWorldQA | POPE |
|---|---|---|---|---|---|---|---|---|
| w/o Cold Start | RL | 79.8 | 80.8 | 94.4 | 693/1707 | 68.5 | 67.1 | 86.0 |
| Cold Start (2K) | Base | 76.4 | 78.7 | 92.4 | 444/1354 | 58.3 | 47.2 | 86.6 |
| | RL | 77.7 | 80.2 | 93.0 | 622/1624 | 62.3 | 52.8 | 86.2 |
| Cold Start (8K) | Base | 76.8 | 78.2 | 90.5 | 525/1368 | 60.5 | 36.5 | 84.8 |
| | RL | 79.2 | 79.4 | 92.5 | 609/1637 | 66.2 | 55.8 | 85.6 |

decide when to answer directly and when to request a high-resolution image, the SFT model exhibits a significantly higher image resize calling ratio across all benchmarks. This behavior is especially evident in the RealWorldQA benchmark, where high-resolution images are generally unnecessary, yet the SFT model still issues requests 62.1% of the time.

Based on this observation, we find that SFT does not enable the model to become "smart" enough to accurately determine whether a high-resolution image is necessary and also requires constructing the training set using GPT-4o. In contrast, RL makes the VLM smarter and more generalizable, and can directly use the original QA pairs without additional formatting.

## C.2 Why not Cold-Start?

Currently, most explorations of RL in VLMs require a cold start stage. In this section, we explore why we do not use a cold start and instead train the model directly with RL.

To investigate this problem, we collect datasets of 2K and 8K samples to cold start our model. The cold-start data are constructed similarly to Sec. C.1, where GPT-4o is used to simulate both the requests for high-resolution images and the corresponding reasoning processes, enabling the SFT training data to closely mimic the behavior of the model trained via RL.

We first compare the performance of the cold-started models before RL training, as shown in the 'Base' lines of Table 13. Although performance improves with increasing data size, the cold-start models still fall short of the original Qwen2.5VL. We believe this is primarily due to the limited diversity and coverage of our data compared to the SFT data used by the Qwen team, resulting in the observed performance gap.

Furthermore, we compare the performance of models after RL training, using the same RL setup as VisionThink, as shown in the 'RL' lines of Table 13. While RL improves the performance of cold-start models, they still fall short compared to models trained from vanilla Qwen2.5VL with RL.

Based on this observation, we conclude that due to the lower diversity and coverage of the cold-start data compared to the original Qwen2.5VL SFT data, introducing cold-start training may improve performance in specific domains covered by the cold-start data but significantly reduces the model's general capability. This limitation restricts its broader applicability. Therefore, in this paper, we do not utilize the cold-start stage.

## C.3 Different Prompt Impact

Since we do not adopt a cold-start stage, designing an appropriate initial prompt becomes crucial to ensure the model begins from a good starting point. The base model Qwen2.5VL-Instruct, which has not undergone RL training, typically tends to answer questions directly and lacks the ability to smartly request high-resolution images when needed.

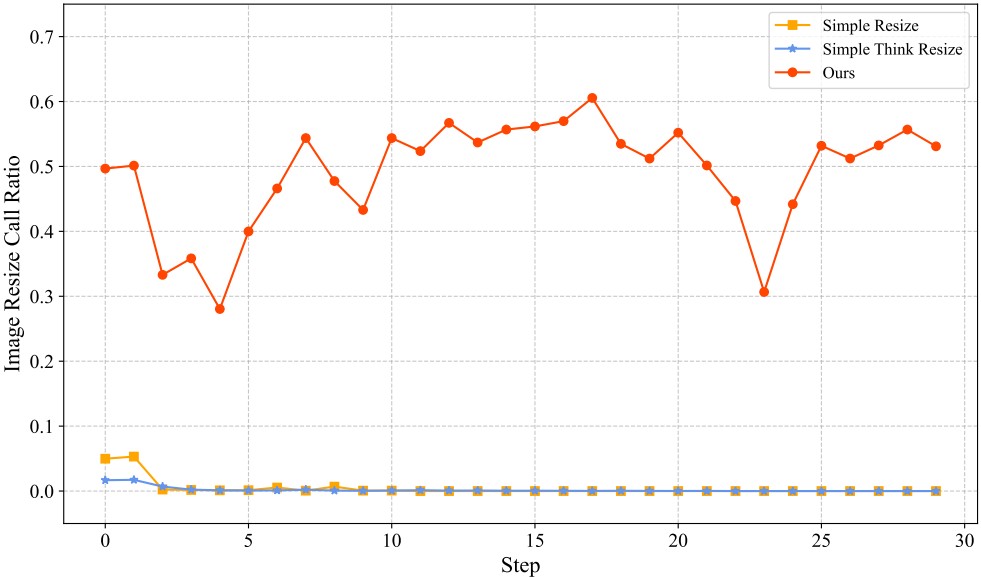

Figure 7: **Impact of Prompt Choice.** Prompts lead to substantial variation in image resize call ratios, with the Qwen official agent prompt demonstrating the most effective performance.

Therefore, it is essential for the base model, when conditioned on our prompt, to show some preference for calling image resizing. Otherwise, if it is overly biased toward direct answering, the GRPO training process will fail to optimize effectively and may collapse into the direct-answering mode. To address this, we compare three prompt settings. The first is the official agent prompt from Qwen's cookbook, shown in Table 4. The other two are our custom-designed prompts, detailed in Table 14.

Table 14: **Two custom prompts for analyzing the impact of different prompts.** The Question placeholder will be replaced with the specific question during training and inference.

| |
|---|
| ***Simple Resize System Prompt:*** |
| You are a helpful assistant. |
| ***Simple Resize User Prompt:*** |
| Answer the user's question based on the image provided. You can place `<resize></resize>` at the end of your response to call the image resize tool, it will return the resized image with its resolution doubled to help you better answer the question. Once you confirm your final answer, place the final answer inside `<answer>` and `</answer>`. Here is the image and question: Question. |
| ***Simple Think Resize System Prompt:*** |
| You are a helpful assistant. Answer the user's question based on the image provided. You can place `<resize>` at the end of your response to call the image resize tool, it will return the resized image with its resolution doubled to help you better answer the question. Once you confirm your final answer, place the final answer inside `<answer>` and `</answer>`. |
| ***Simple Think Resize User Prompt:*** |
| Enclose reasoning in `<think></think>` Question. |

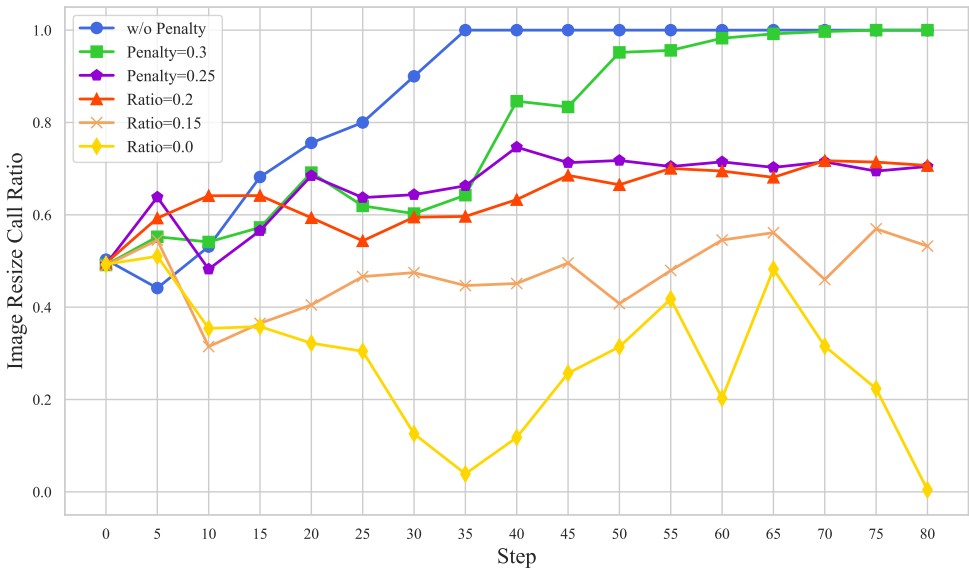

Figure 8: **Ablation Study on Penalty Ratio Threshold.** As the threshold increases, the model progressively favors requesting image resizing instead of providing direct answers.

As shown in Fig. 7, our model using the official agent prompt demonstrates a higher image resize call ratio on the effective batch, which consists of an equal mix of high-resolution-required samples and direct-answer samples. In contrast, the other two custom-designed prompts perform poorly, both in their initial behavior and in the final trained model's ability to correctly request image resizing, and ultimately collapse into consistently producing direct answers. Based on these analyses, we find that the Qwen official agent prompt, likely optimized during the pretraining or supervised fine-tuning stages by the Qwen team, is more suitable for VisionThink.

## C.4 Ablation Study on Penalty Control Threshold

In the main paper Eq. 5, we design the penalty ratio as below:

$$\mathcal{P}_{control} = 0.1 \cdot \left[\mathbf{1}_{\text{direct}}\mathbb{I}(r < \theta) + \mathbf{1}_{\text{high}}\mathbb{I}(r \geq \theta)\right], \qquad r = \frac{C_{\text{direct}}}{C_{\text{direct}} + C_{\text{high}}}, \qquad (6)$$

where $\theta$ is the threshold.

Intuitively, the larger the value of $\theta$, the more likely the model is to penalize direct answers, thereby encouraging it to request high-resolution images. Conversely, a smaller $\theta$ leads the model to penalize responses that call for image resizing, thus promoting direct answers. Based on this intuition, we experimented with different threshold values and recorded the proportion of high-resolution image requests within the effective batch. The results are shown in Fig. 8. As indicated by the Eq. 6, increasing the threshold gradually shifts the model's behavior from favoring direct answers to favoring image resizing requests. Eventually, the model collapses into always requesting high-resolution images. However, within an appropriate range, the model's behavior is not highly sensitive to the exact threshold value.

Besides, as shown in the Table. 15, we report the performance and inference time of models trained with different penalty ratios. Overall, the trends are consistent with Fig. 8: as the penalty ratio increases, the model becomes more inclined to resize the input image, leading to higher inference time.

In general VQA scenarios, model performance does not significantly improve with increased inference time. However, for fine-grained understanding scenarios (e.g., ChartQA, OCRBench), increasing the penalty ratio encourages more image resizing, which results in both longer inference time and improved performance.

Table 15: **Benchmark performance under different penalty ratios.** Perf denotes the model performance, and Time indicates the real-world inference time.

| Penalty | ChartQA | | OCRBench | | MathVista | | MME | | MMMU | | RealWorldQA | |
|---------|---------|----------|----------|----------|-----------|----------|------|----------|------|----------|-------------|----------|
| | Perf | Time (s) | Perf | Time (s) | Perf | Time (s) | Perf | Time (s) | Perf | Time (s) | Perf | Time (s) |
| ratio=0 | 63.0 | 353.2 | 70.3 | 146.0 | 65.6 | 1097.1 | 2272 | 405.5 | 50.1 | 308.5 | 64.8 | 112.3 |
| ratio=0.15 | 74.1 | 629.5 | 75.8 | 184.7 | 65.7 | 1306.5 | 2347 | 465.2 | 49.4 | 504.7 | 66.8 | 198.4 |
| ratio=0.2 | 79.8 | 746.1 | 80.8 | 211.8 | 67.5 | 1745.5 | 2400 | 653.0 | 51.2 | 608.4 | 67.1 | 235.6 |
| ratio=0.25 | 80.2 | 725.0 | 83.5 | 425.5 | 67.9 | 1334.6 | 2278 | 938.3 | 50.7 | 570.0 | 67.5 | 361.9 |
| ratio=0.3 | 81.0 | 1033.4 | 84.8 | 435.6 | 66.6 | 1666.0 | 2235 | 1932.7 | 49.0 | 866.9 | 69.3 | 492.8 |
| w/o Penalty | 82.1 | 1093.6 | 85.3 | 437.1 | 67.5 | 1721.9 | 2354 | 1843.8 | 50.7 | 820.5 | 68.0 | 436.1 |

Table 16: **Bias influence of LLM-as-Judge.** We compare the results of several LLM models.

| Model | MMMU | MMMU-Pro | MMBench | RealWorldQA | POPE | MME | MathVista | MMVet |
|-------|------|----------|---------|-------------|------|-----|-----------|-------|
| GPT-4o | 52.7 | 41.1 | 83.4 | 66.5 | 88.6 | 2314 | 71.2 | 69.5 |
| Qwen2.5-72B-Instruct | 52.6 | 40.2 | 84.2 | 66.1 | 88.4 | 2360 | 70.3 | 69.1 |
| Qwen2.5-3B-Instruct | 51.9 | 38.5 | 82.6 | 66.9 | 87.7 | 2379 | 70.6 | 68.9 |
| Qwen3-1.7B | 51.8 | 38.1 | 82.8 | 67.7 | 87.9 | 2210 | 69.1 | 66.8 |

## C.5 The Bias Influence of LLM-as-Judge

VisionThink employs an LLM-as-Judge paradigm to assess textual answers in open-ended VQA tasks. This naturally raises a question: **does the inherent bias of the LLM influence the evaluation results?** To mitigate potential biases, we adopt the following three design strategies:

**(1) Using the LLM-as-Judge not the VLM-as-Judge.** Since LLMs generally have stronger capabilities than VLMs, employing an LLM reduces hallucinations and improves reliability.

**(2) Filtering the dataset.** We filter out subjective open-ended questions that have multiple valid answers, such as image descriptions. The remaining questions have clear ground truth, e.g., Q: <image> Who is the author of this book? A: Dewey Lambdin.

**(3) Carefully designing the prompt.** Our prompt requires the LLM to return a discrete value: 1 for a correct answer and 0 for an incorrect one, instead of a continuous score. This binary format minimizes ambiguity and reduces the chance of misjudgment.

Furthermore, as shown in the Table 16, we conduct additional experiments to further investigate this issue. Specifically, we compare the judgments made by GPT-4o and Qwen2.5-72B with those from smaller models such as Qwen2.5-3B and Qwen3-1.7B. While larger models achieve slightly better performance, the smallest model, Qwen3-1.7B, still achieves comparable results under our carefully designed setup. This indicates that LLM model bias has limited influence on our VisionThink.

## C.6 Discussion about Zoom-in Strategy

With the release of GPT-O4, many researchers have begun to focus on zoom-in strategies. Therefore, in this section, we provide a brief discussion on this topic.

Zoom-in methods rely heavily on strong localization capabilities, which current models such as Qwen2.5VL-7B do not yet fully possess. Inaccurate localization may lead to suboptimal region selection, thereby degrading overall performance. Moreover, zoom-in strategies often require multiple iterative refinement steps due to imperfect region proposals, which can result in even higher total token consumption compared to our approach.

Nevertheless, we acknowledge that this is a highly promising research direction and have already included it as part of our future work.

# D  Qualitative Results

In this section, we present a case study comparing our proposed VisionThink with other efficient VLM methods: FastV [8], SparseVLM [100], and VisionZip [80]. As shown in the three cases below, for OCR-related or detail-intensive samples, our proposed VisionThink model can smartly determine when a high-resolution image is needed. In contrast to previous efficient VLMs, which suffer performance degradation due to fixed reduction ratios, VisionThink avoids such issues by making adaptive decisions based on the input, thereby maintaining strong performance.

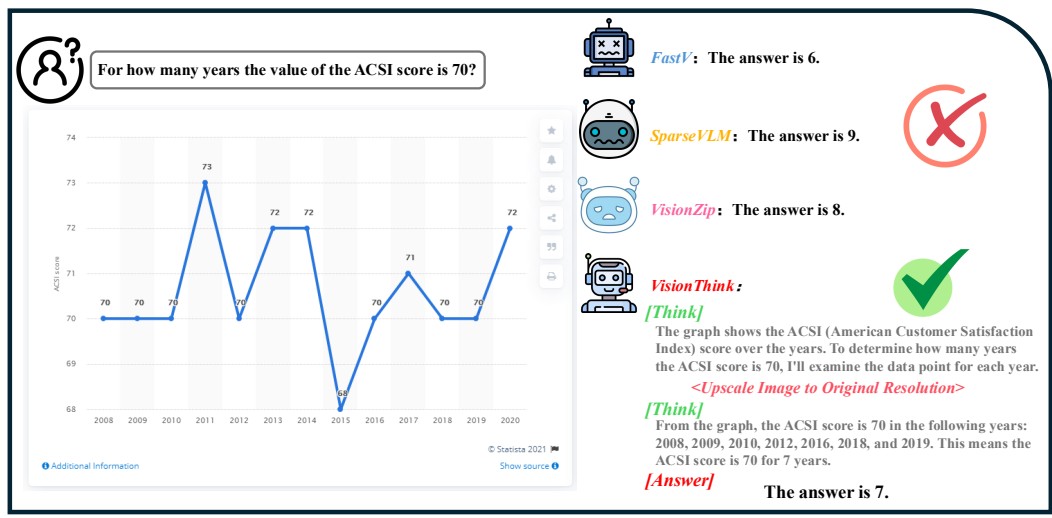

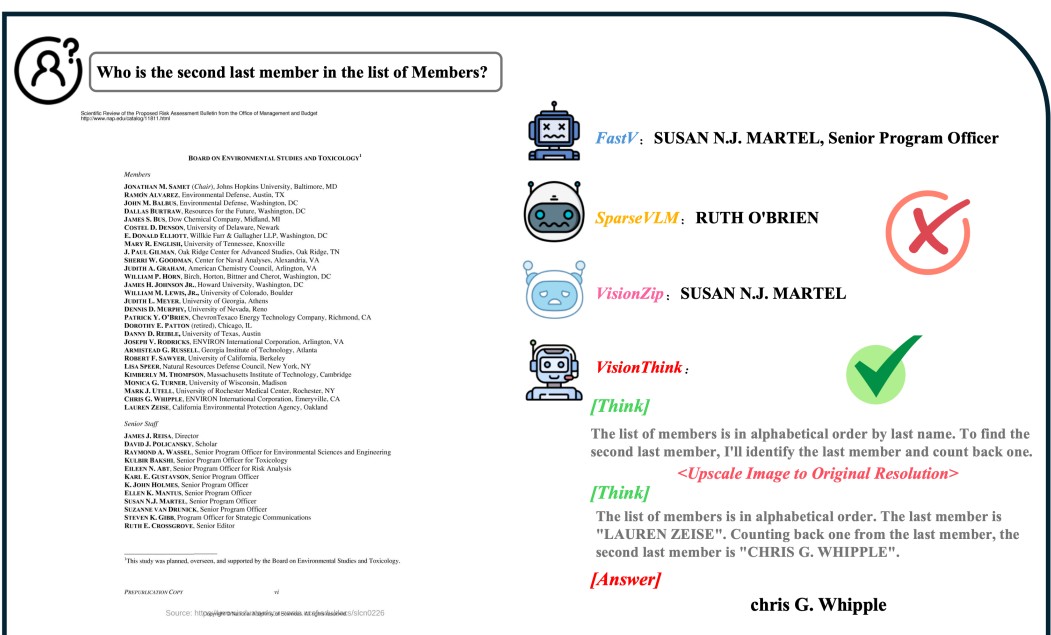

# E  Broader Impact Statement

This paper is conducted solely for academic research purposes. All data used in this work were collected through compliant and ethical channels, ensuring adherence to relevant data protection and usage guidelines. Furthermore, all models employed in this study comply with their respective

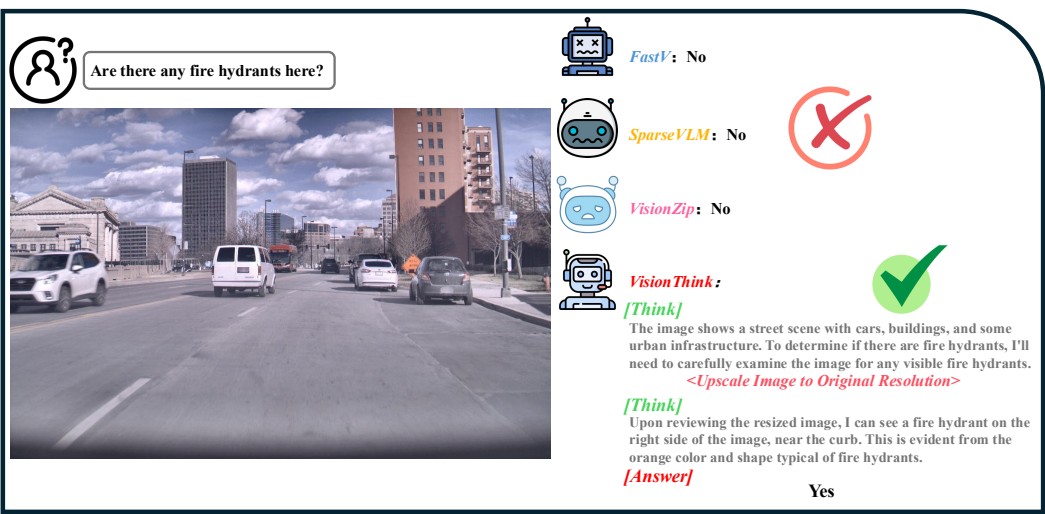

license agreements. As such, this research upholds high standards of integrity and responsibility, with no foreseeable negative societal impact.

