# OpenReview forum: "VisionThink: Smart and Efficient Vision Language Model via Reinforcement Learning"
_NeurIPS.cc/2025/Conference — NeurIPS 2025 poster_

### Official Review · Reviewer_Ubhv · 2025-06-18

**Clarity:** 3
**Significance:** 2
**Originality:** 3
**Rating:** 4
**Confidence:** 4

**Summary:**

This paper introduces VisionThink, a novel paradigm to improve both efficiency and performance by dynamically adjusting image resolution. The core idea is to initially process a downsampled image and, if necessary, autonomously request a higher-resolution image. To achieve this, VisionThink combines reinforcement learning and LLM-as-Judge strategy for effective training on general VQAtasks. A reward function with a penalty mechanism prevents the model from always requesting high-resolution images.

**Questions:**

- Figure 5 shows that OCRBench has over one-third of its questions directly answered, yet Figure 4 demonstrates very little acceleration effect for OCRBench. What's the reason behind this discrepancy?
- The "LLM as eval combined with RL" strategy frequently exhibits completely opposite performance between full-image resolution and mixed-image resolution. Could you elaborate on this issue?
- Although the paper ablates the impact of different penalties on the Image Resize Call Ratio, it doesn't show specific performance changes. Is the model sensitive to penalty settings? How do different penalties affect the results on various benchmarks?

**I would be happy to raise my score if these weaknesses and questions can be addressed.**

**Ethical Concerns:**

["NO or VERY MINOR ethics concerns only"]

**Final Justification:**

Thank you for the authors' dedicated response. I am pleased to see work that introduces Reinforcement Learning (RL) to token acceleration. However, this implies that RL needs to be re-applied for different models. Additionally, considerations about the most appropriate scale for resizing and whether the process can be extended from two to multiple times are necessary. The issues that arise with the method are ambiguous, I will maintain my original score of Borderline accept.

**Limitations:**

yes

**Paper Formatting Concerns:**

There is no major formatting issue.

**Quality:**

3

**Strengths And Weaknesses:**

**Strengths**
- Unlike previous efficient VLM methods that rely on fixed pruning ratios or thresholds, VisionThink autonomously decides whether to compress tokens on a case-by-case basis.
- The extensive experiments show that VisionThink achieves substantial savings in inference time on most benchmarks.

**Weaknesses**
- Similar to how RL combined with additional image processing can be used for "zoom in" operations to introduce fewer visual tokens, it's necessary to explain the advantages of upsampling compared to zooming in.
- The impact of benchmark characteristics on the method needs further specific analysis. For example, how would the acceleration effect and performance change if the benchmark inputs tend to be high-resolution or low-resolution images. These can help further understanding of the method.
- For fine-grained understanding where the original image resolution is already quite large, the acceleration effect may be weaker because the additional inference overhead from an extra inference step is relatively high.
- Although the acceleration effect compared to Qwen-RL has been demonstrated, further comparison of the inference efficiency with other efficient VLMs is also needed.

---

> ### Author Rebuttal · Authors · 2025-07-30
>
> Dear Reviewer Ubhv,
>
> Many thanks for your constructive and insightful feedback! We are glad that you found our VisionThink propose a novel paradigm. We have revised the paper following your suggestions, and will address both your questions and weaknesses below.
>
> ---
> ## Questions
>
> > ### [Q1] Discrepancy in Time Acceleration on OCRBench
> Thank you for the insightful question. We clarify this from two aspects:
>
> 1. On OCRBench, the upper bound for efficiency is `Qwen-RL (1/4)`, which always uses a 1/4-resolution image, resulting in roughly **50% time savings** compared to full-resolution input. In contrast, our method uses 1/4 resolution in **~38%** of the cases and requests high-resolution input for the remaining **~62%**. Achieving a **\~20% overall time reduction** under this mixed policy is therefore consistent and reasonable.
>
> 2. Additionally, we emphasize that our reported time is **real-world GPU runtime**, not theoretical FLOPs or token counts. As a result, this may introduce variance from multiple system-level factors, but it more accurately reflects practical deployment scenarios.
>
> All evaluation code and measurement procedures will be released to ensure reproducibility. We welcome continued community oversight.
>
>
> > ### [Q2] Opposite performance between full-image resolution and mixed-image resolution.
>
> Thank you for the insightful question. In a few benchmarks, we observe that the mixed-resolution outperforms the full-resolution setting. We believe this result arises from two key factors:
>
> 1. For certain samples requiring resize, the model processes the same image twice at different resolutions, effectively gaining a dual-view perspective. This may serve as a form of **implicit data augmentation**, yielding marginal performance gains.
>
> 2. More importantly, the **diverse sampling during RL training** enables the model to explore a broader policy space. This exploration may occasionally yield better policies, especially on specific benchmarks.
>
> We will clarify this observation in the final version. Thanks again for your point it.
>
> > ### [Q3.1] Model sensitivity to the penalty settings
>
> Thanks for your professional questions, and we have discussed this problem in the *Appendix Sec C.4 Ablation Study on Penalty Control Threshold*. The results show that, within an appropriate range, the model’s behavior is not highly sensitive to the
> exact threshold value.
>
> Thanks again for your point it out, we will consider move this part into the main paper to make it clearly.
>
>
> > ### [Q3.2] How do different penalties affect the results on various benchmarks?
>
> In *Appendix Figure 3*, we show how the penalty influences model behavior. Here, we further provide quantitative results across various benchmarks.
>
> As shown in the table below, we report the performance and inference time of models trained with different penalty ratios. Overall, the trends are consistent with *Appendix Figure 3*: as the penalty ratio increases, the model becomes more inclined to resize the input image, leading to higher inference time.
>
> In **general VQA scenarios**, model performance does not significantly improve with increased inference time. However, for **fine-grained understanding scenarios** (e.g., *ChartQA*, *OCRBench*), increasing the penalty ratio encourages more image resizing, which results in both longer inference time and improved performance.
>
> | Penalty   | ChartQA Performance | ChartQA Time | OCRBench Performance | OCRBench Time | MathVista Performance | MathVista Time | MME Performance | MME Time | MMMU Performance | MMMU Time | RealWorldQA Performance | RealWorldQA Time |
> |--------------|---------|--------|-----|--------|-----------|--------|-----|--------|------|-------|------|---------|
> | ratio=0      | 63.0    | 353.2s  | 70.3| 146.0s  | 65.6      | 1097.1s |2272 | 405.5s  |50.1 | 308.5s |64.8 | 112.3s   |
> | ratio=0.15   | 74.1    | 629.5s  | 75.8| 184.7s  | 65.7      | 1306.5s |2347 | 465.2s  |49.4 | 504.7s |66.8  | 198.4s   |
> | ratio=0.2    | 79.8    | 746.1s  | 80.8| 211.8s  | 67.5      | 1745.5s |2400 | 653.0s  |51.2  | 608.4s |67.1  | 235.6s   |
> | ratio=0.25   | 80.2    | 725.0s  | 83.5| 425.5s  | 67.9      | 1334.6s |2278 | 938.3s  |50.7 | 570.0s |67.5 | 361.9s   |
> | ratio=0.3    | 81.0    | 1033.4s | 84.8| 435.6s  | 66.6      | 1666.0s |2235 | 1932.7s |49.0  | 866.9s |69.3 | 492.8s   |
> |w / o Penalty | 82.1    | 1093.6s | 85.3| 437.1s  | 67.5      | 1721.9s |2354 | 1843.8s |50.7 | 820.5s |68.0 | 436.1s   |
>
> Thank you for your constructive question. We will include these results in the paper, as we believe they significantly enhance the completeness and depth of our work.
>
>
>
> ---
> ## Weaknesses
>
> > ### [W1] Advantage comparison to zoom-in strategy
> Thank you for the professional question. We see the advantages of our approach over zoom-in strategies from two key perspectives:
>
> 1. Zoom-in methods require **strong localization capabilities**, which current models like Qwen2.5VL-7B do not fully possess. Inaccurate localization can degrade performance.
>
> 2. Zoom-in may require multiple iterative steps due to imprecise region proposals, potentially leading to **higher total token consumption** than our strategy.
>
> However, we also agree this is a very promising direction and have already noted it as part of our future work (Line 285). We will incorporate this discussion into the camera-ready version to better contextualize the comparison.
>
> > ### [W2] Impact of benchmark characteristics on method behavior
>
> **Settings.** We selected the OCR-related benchmark *ChartQA*, along with two benchmarks representing more general VQA scenarios: *MME* and *RealWorldQA*. For each, we computed the ratio of samples that triggered image resizing under different input token ranges.
>
> **Results.**
> As shown in the table below, there is no clear correlation between image size (token range) and the likelihood of image resizing. Instead, the **task type itself** has a stronger influence on the model’s behavior. For example, *ChartQA* consistently exhibits higher image-resize ratios across all ranges, whereas *RealWorldQA* remains low. This suggests that **task characteristics**, rather than resolution alone, play a dominant role in guiding VisionThink’s resizing decisions.
>
> Thank you again for your suggestion. We will include this analysis in the camera-ready version of the paper to make our discussion more complete.
>
> | Token Range | MME (%) | RealWorldQA (%) | ChartQA (%) |
> | ----------- | ------- | --------------- | ----------- |
> | 0–600       | 32.8    | 8.0             | 84.0        |
> | 600–900     | 26.0    | 5.0             | 63.0        |
> | 900–1 500   | 39.1    | 19.7            | 51.1        |
> | >1 500      | 21.3    | 32.4            | 61.5        |
>
>
> > ### [W3] Efficiency on high-resolution images
>
> Thanks for pointing this out. Following your suggestion, we conducted further analysis.
> As shown in the table below, we focus on DocVQA, the benchmark with the highest image resolution. Our results show that VisionThink achieves **significant acceleration** while maintaining **comparable accuracy.**
>
> We believe the reason is that, for high-resolution images, even after resizing (e.g., halving both height and width), the image remains **sufficiently clear** to solve most questions — even on fine-grained tasks like DocVQA.
>
> We will clarify this point in the camera-ready version. Thank you again for the helpful suggestion.
>
> | Task | mean resolution | metric |Qwen-RL | Qwen-RL (1/4) | VisionThink |
> | ----------- | ------- | ------ | --------- | ----------- | ----------- |
> | DocVQA   | 2099 $\times$ 1783 (H$\times$W) | Accuracy    | 95.1            | 94.3        | 94.4 |
> |       |   | Time | 3076s            | 1327s        | 1355s |
>
>
> > ### [W4] Further comparison with additional efficient VLMs
>
> Thank you for the valuable suggestion. In *Appendix B.6 (Comparison with Previous Efficient VLMs)*, we have included additional comparisons with recent state-of-the-art efficient VLMs.
> We appreciate your reminder and will move this section into the main paper in the camera-ready version to make the comparison more prominent and accessible to readers.
>
> ---
> **Thank you again for helping us improve the paper and hope our response can resolve your concerns! Please let us know if you have any further questions. We will be actively available until the end of rebuttal period. If you feel your concerns are addressed, please consider reevaluating our work. Looking forward to hearing from you :-) !**

---

> > ### Comment · Reviewer_Ubhv · 2025-08-05
> >
> > Thank you to the author for the detailed response。 some of my questions have now been resolved. However, I still have some doubts regarding Q1 and Q3.2:
> >
> > **Q1:**
> > 1. OCRBench and MME have a nearly identical proportion of "direct answers," yet their speedup ratios are completely different. The authors may need to explain this by comparing the data characteristics of the different benchmarks.
> >
> > 2. Additionally, on ChartQA, nearly 30% of questions are answered directly, yet the inference cost of VisionThink is, conversely, higher than that of Qwen-RL.
> >
> > 3. Qwen-VL (1/4) typically has a 1.5x-2x speedup in terms of inference cost compared to VisionThink. Perhaps for a fairer comparison, the resolution should be increased to ensure a similar inference cost.
> >
> > **Q3.2**:
> > 1. On the MathVista benchmark, the inference cost fluctuates as penalty changes, unlike on other benchmarks where it tends to first increase and then decrease. Could you please provide an explanation for this?

---

> > > ### Author Response · Authors · 2025-08-05
> > > **Response to additional questions (Q1)**
> > >
> > > Dear Reviewer Ubhv,
> > >
> > > Thank you for your kind response! We’re glad to hear that some of your concerns have been addressed. Your professional feedback has truly helped strengthen our paper. Below, we respond to your remaining questions:
> > >
> > > ---
> > >
> > > > ### Q1.1:  OCRBench and MME have a nearly identical proportion of "direct answers," yet their speedup ratios are completely different.
> > >
> > > We'd like to clarify that the proportions of "direct answers" shown in main paper Figure 5 are actually **not identical**:
> > >
> > > * For **OCRBench**, the **direct answer ratio is 38%**, while **"Apply Resize"** accounts for **62%**.
> > > * In contrast, for **MME**, the **"Apply Resize"** ratio is only **31%**, with **69% being direct answers**.
> > >
> > > So the distributions are **nearly reversed** between the two benchmarks. This **aligns** with the observed **speedup differences**:
> > >
> > > * On **OCRBench**, since most samples require high-resolution input, the speedup is limited.
> > > * On **MME**, where most inputs are handled directly, the speedup is more substantial.
> > >
> > > We **fully understand** how the figure may have caused confusion. The color similarity likely contributed to the misinterpretation. We will improve the visualization and clarify this point in the revised version.
> > >
> > >
> > > Thanks again for catching this!
> > >
> > >
> > >
> > > > ### Q1.2: ChartQA inference cost higher than that of Qwen-RL.
> > >
> > >
> > > Thank you for your insightful question! We will clarify it explicitly in the camera-ready version.
> > >
> > > On **ChartQA**, our VisionThink policy uses **21% direct answers** (consuming 25% of tokens) and **79% Apply Resize** (requiring two rounds: 25% + 100% = 125% tokens).
> > > A rough estimate of the overall token usage is:
> > >
> > > > 0.21 × 0.25 + 0.79 × 1.25 ≈ 1.03
> > >
> > > which is **slightly higher than Qwen-RL (100%)**.
> > >
> > > Additionally, real-world inference time is subject to variance from **system-level and infrastructure factors**, such as batch handling, memory reuse, and communication overhead in multi-round settings.
> > >
> > > Therefore, it is reasonable that VisionThink incurs a slightly higher inference cost than Qwen-RL on ChartQA.
> > >
> > >
> > >
> > > > ### Q1.3: Increasing the baseline resolution for Qwen-RL (1/4)
> > >
> > > Thank you for this very insightful suggestion, we sincerely appreciate it!
> > >
> > > To address your concern, we increased the input image resolution for Qwen-RL (1/4) on several benchmarks individually, ensuring that its inference time is comparable to or slightly higher than VisionThink. The results are shown below:
> > >
> > > | Method             | ChartQA | OCRBench | MathVista | MME   | MMMU  | RealWorldQA | Avg.   |
> > > | ------------------ | ------- | -------- | --------- | ----- | ----- | ----------- | --------- |
> > > | **Qwen-RL (1/4)↑** | 75.4    | 77.2     | 67.8      | 2175  | 51.2  | 65.3        | 69.10     |
> > > | Time (s)           | 676.4   | 232.5    | 1792.3    | 703.2 | 630.8 | 258.1       | 715.6     |
> > > | **VisionThink**    | 79.8    | 80.8     | 67.5      | 2400  | 51.2  | 67.1        | 72.69 |
> > > | Time (s)           | 746.1   | 211.8    | 1745.5    | 653.0 | 608.4 | 235.6       | 700.1 |
> > >
> > > As shown, even after increasing the resolution for Qwen-RL (1/4), its performance still falls short of VisionThink, **especially on OCR-related tasks** like ChartQA and OCRBench.
> > >
> > > Additionally, we will include a new **Qwen-RL (1/2)** baseline (RL-trained with half-resolution images) in the camera-ready version to further strengthen the comparison and support the effectiveness of VisionThink.
> > >
> > > We sincerely appreciate your understanding, especially as the discussion period is drawing to a close.
> > >
> > > ---
> > >
> > > Beyond this specific evaluation, your insightful question inspired us to have a broader reflection:
> > > **What does VisionThink offer the community beyond just performance and efficiency gains?**
> > >
> > >
> > > Now, in our view, **VisionThink may offer two key conceptual contributions**:
> > >
> > > * **A new image reading paradigm for VLMs.**
> > >    VisionThink represents a new paradigm for image reading, distinct from traditional VLMs. It leverages Reinforcement Learning to train models to think with images, enabling an automatic coarse-to-fine visual reasoning strategy. This framework naturally extends to tasks like image zoom-in and video token selection, offering **a smarter way and more adaptive way to process visual input.**
> > >
> > >
> > > * **A step of exploration toward tool-augmented VLMs.**
> > >    VisionThink explores learned decision-making over when and how to use visual resources (e.g., resize), aligning with the broader vision of **VLM-as-agent** settings. We hope our penalty-based mechanism could also offer a little insight for future work on tool use in multimodal systems.
> > >
> > > ---
> > >
> > > **We sincerely thank you for your insightful questions and discussions, which have been equally inspiring to us.** We will incorporate these discussions into the final version of the paper.

---

> > > ### Author Response · Authors · 2025-08-05
> > > **Response to additional questions (Q3.2)**
> > >
> > > > ### Q3.2: Inference cost fluctuates as penalty changes
> > >
> > > Thank you for carefully reviewing our rebuttal, and we truly appreciate your detailed analysis. Let us further discuss on this point.
> > >
> > > ---
> > >
> > > #### **(1) Clarification on the Penalty–Performance–Time Relationship**
> > >
> > > You mentioned that,
> > > > **unlike on other benchmarks where it tends to first increase and then decrease.**
> > >
> > > We appreciate your observation and would like to **offer a different perspective**, supported by the following table:
> > >
> > >
> > > | Penalty       | Total Time (s)       | Avg Performance (%) |
> > > | -------------  | ---- | ------------------- |
> > > | ratio=0       | 353.2 + 146.0 + 1097.1 + 405.5 + 308.5 + 112.3 = **2422.6s**   | (63.0 + 70.3 + 65.6 + 2272/2800 + 50.1 + 64.8) / 6 = **65.49** |
> > > | ratio=0.15    | 629.5 + 184.7 + 1306.5 + 465.2 + 504.7 + 198.4 = **3288.9s**   | (74.1 + 75.8 + 65.7 + 2347/2800 + 49.4 + 66.8) / 6 =  **69.94** |
> > > | ratio=0.2     | 746.1 + 211.8 + 1745.5 + 653.0 + 608.4 + 235.6 = **4200.4s**   | (79.8 + 80.8 + 67.5 + 2400/2800 + 51.2 + 67.1) / 6 = **72.69** |
> > > | ratio=0.25    | 725.0 + 425.5 + 1334.6 + 938.3 + 570.0 + 361.9 = **4355.3s**   | (80.2 + 83.5 + 67.9 + 2278/2800 + 50.7 + 67.5) / 6 = **71.86** |
> > > | ratio=0.3     | 1033.4 + 435.6 + 1666.0 + 1932.7 + 866.9 + 492.8 = **6427.4s** | (81.0 + 84.8 + 66.6 + 2235/2800 + 49.0 + 69.3) / 6 = **71.42** |
> > > | w / o Penalty | 1093.6 + 437.1 + 1721.9 + 1843.8 + 820.5 + 436.1 = **6352.9s** | (82.1 + 85.3 + 67.5 + 2354/2800 + 50.7 + 68.0) / 6 = **72.94** |
> > >
> > > (Performance calculated as average over above rebuttal benchmarks, with MME normalized to 100)
> > >
> > >
> > >
> > > We suggest viewing the results in **three groups**:
> > >
> > > ---
> > >
> > > **Group1: `ratio=0`**
> > >
> > > As shown in *Appendix Figure 3*, the model collapses into **rarely calling `Resize Image`**. Consequently, as reflected in the table above, its performance, particularly on OCR-heavy benchmarks like **ChartQA** and **OCRBench**, is clearly suboptimal.
> > >
> > > ---
> > >
> > > **Group2: `ratio=0.15`, `0.2`, `0.25`**
> > >
> > > This range reflects **balanced behavior**, where the model learns to smartly call `Resize Image` only when necessary.
> > >
> > > * In *Appendix Figure 3*, the `Resize Image` ratio remains moderate.
> > > * `ratio=0.15` achieves **lower inference time**, but its performance is slightly worse than `ratio=0.2` and `0.25`,
> > >   reflecting a classic **efficiency–effectiveness trade-off**.
> > >   Our recommended VisionThink configuration falls within this range.
> > > * Moreover, as shown in the table, although this group has higher inference time than `ratio=0`, it achieves **substantial performance improvement** in return.
> > >
> > > ---
> > >
> > > **Group3: `ratio=0.3` and `w/o Penalty`**
> > >
> > > In these two settings, the model **collapses into always calling `Resize Image`**, as observed in *Appendix Figure 3*.
> > > * This results in **significantly higher inference time**.
> > > * However, across both general and fine-grained VQA tasks, the performance gains over **Group 2** are **negligible**.
> > >
> > > ---
> > >
> > > **Summary**
> > >
> > > The trends across the table and *Appendix Figure 3* are **consistent**:
> > >
> > > * As the penalty increases, **both performance and inference time initially improve**.
> > > * Beyond a point, **inference time keeps rising**, but **performance shows no further gain**.
> > >
> > >
> > > This also supports the design goal of VisionThink: enabling the model to automatically trigger image resize when it recognizes insufficient information, thereby maximizing efficiency without sacrificing performance.
> > >
> > > ---
> > >
> > > #### **(2) On MathVista Behavior**
> > >
> > > We believe the fluctuation in inference cost on MathVista as the penalty changes is primarily due to two factors:
> > >
> > >
> > > * MathVista is a challenging reasoning benchmark. On `lmms-eval` evaluation process, one of its subsets, **MathVista-COT**, requires **CoT-formatted answers**, which is rare among existing benchmarks. This results in variable output lengths and variations in decoding time.
> > >
> > > * **More importantly**, unlike SFT, **RL training does not imitate ground truth**. Instead, the model learns via **diverse sampling**, which enables exploration of a **broader policy space**. This diversity can lead to **variance in performance and behavior across individual benchmarks**.
> > >
> > >
> > > Given these factors, we **strongly recommend** evaluating performance and efficiency, especially for **RL-trained models**, from an **overall multi-benchmark perspective** (as discussed in (1) Clarification on the Penalty–Performance–Time Relationship), rather than based on individual tasks.
> > >
> > >
> > > Thank you again for raising this important point. We will incorporate this discussion in the camera-ready version.
> > >
> > > ---
> > >
> > > **We truly appreciate your thoughtful questions and the time you took to engage with us — they’ve inspired us and given us much to reflect on.** We will incorporate these discussions into the final version of the paper.
> > >
> > > If you have any further questions, we’d be happy to address them. If you feel your concerns have been fully addressed, we’d greatly appreciate your reconsideration of the overall evaluation.
> > >
> > > Thank you once again!

---

> > > > ### Comment · Reviewer_Ubhv · 2025-08-06
> > > >
> > > > Thank you for providing all the addtional results. Most of my questions have been resolved. I look forward to seeing the model's performance with Qwen-RL(1/2) maintaining the same inference time in a future version. This experiment is very important for demonstrating the effectiveness of VisionThink, but I also understand the time constraints for discussion.

---

> > > > > ### Author Response · Authors · 2025-08-07
> > > > > **Additional comparison results**
> > > > >
> > > > > Dear Reviewer Ubhv,
> > > > >
> > > > > We’re glad to hear that most of your concerns have been addressed, and we sincerely thank you for your patience throughout this process. To run these additional experiments, we borrowed several GPUs, and we’re happy to now share the results with you!
> > > > >
> > > > > We have added **Qwen-RL (1/2)** to the comparison. The results are shown below:
> > > > >
> > > > > | Method             | ChartQA | OCRBench | MathVista | MME   | MMMU  | RealWorldQA | Avg.      |
> > > > > | ------------------ | ------- | -------- | --------- | ----- | ----- | ----------- | --------- |
> > > > > | **Qwen-RL (1/4)↑** | 75.4    | 77.2     | 67.8      | 2175  | 51.2  | 65.3        | 69.10     |
> > > > > | Time (s)           | 646.1   | 192.8    | 1792.5    | 703.0 | 630.4 | 258.1       | 717.1     |
> > > > > | **Qwen-RL (1/2)**  | 76.2    | 76.9     | 66.7      | 2292  | 51.7  | 66.9        | **70.00**     |
> > > > > | Time (s)           | 415.2   | 205.1    | 1946.7    | 731.4 | 667.4 | 362.9       | 721.4     |
> > > > > | **VisionThink**    | 79.8    | 80.8     | 67.5      | 2400  | 51.2  | 67.1        | **`72.69`** |
> > > > > | Time (s)           | 746.1   | 211.8    | 1745.5    | 653.0 | 608.4 | 235.6       | 700.1 |
> > > > >
> > > > >
> > > > > As shown above, **VisionThink achieves better performance** than Qwen-RL (1/2), especially on **OCR-related tasks** such as ChartQA and OCRBench, while also consuming slightly less inference time.
> > > > >
> > > > >
> > > > > Furthermore, on **DocVQA**, **VisionThink** achieves a **significant efficiency gain** and slightly better performance compared to Qwen-RL (1/2).
> > > > >
> > > > >
> > > > > | Task   | Metric   | Qwen-RL (1/2) | VisionThink |
> > > > > | ------ | -------- | ------------- | ----------- |
> > > > > | DocVQA | Accuracy | 94.2          | 94.4        |
> > > > > |        | Time (s) | **2157**          | **`1355`**    |
> > > > >
> > > > > ---
> > > > >
> > > > > Thank you again for your insightful questions. They have helped us strengthen our analysis, and we will incorporate this discussion into the revised version of the paper.

---

> > > > > > ### Comment · Reviewer_Ubhv · 2025-08-08
> > > > > >
> > > > > > Thank you for the supplementary results. There might be an error on ChartQA, as RL(1/2) shows a greater speedup than RL(1/4). Additionally, given that RL(1/4) represents the performance lower bound for VisionThink and RL(1/2) uses an increased image resolution, the performance trend between RL(1/2) and VisionThink should be consistent with that of RL(1/4) and VisionThink. However, the results on OCRBench and MathVista show the opposite trend. I hope more explanation for these points can be provided in the future.

---

> > > > > > > ### Author Response · Authors · 2025-08-08
> > > > > > > **Further clarification**
> > > > > > >
> > > > > > > Dear Reviewer Ubhv,
> > > > > > >
> > > > > > > Thank you for your reply. Please allow me to further clarify the confusion.
> > > > > > >
> > > > > > >
> > > > > > > **Qwen-RL (1/4)↑** refers to the model *trained* on 1/4-resolution inputs but *evaluated* with **upscaled resolution**, in order to ensure its inference time is comparable to or slightly higher than that of VisionThink.
> > > > > > > Therefore, **Qwen-RL (1/4)↑ consumes significantly more tokens** during evaluation and should **not be considered a lower bound**. Specifically, we controlled the token usage by **manually adjusting the input image size** during evaluation.
> > > > > > >
> > > > > > > > **Note**: Qwen-RL (1/4)↑ was introduced during the discussion phase in response to your suggestion to test with higher resolution. At the time, we were concerned that training Qwen-RL (1/2) from scratch might not be feasible due to limited time and compute. Therefore, we used Qwen-RL (1/4)↑ as a temporary solution to demonstrate the effectiveness of **VisionThink**.
> > > > > > >
> > > > > > >
> > > > > > >
> > > > > > > As shown in the table below (focusing on ChartQA, OCRBench, and MathVista as you mentioned), we report both the **token usage** and **inference time** for each variant:
> > > > > > >
> > > > > > >
> > > > > > >
> > > > > > >
> > > > > > > | Metric                    | ChartQA | OCRBench | MathVista |
> > > > > > > | ------------------------- | ------- | -------- | --------- |
> > > > > > > | **Qwen-RL (1/4)↑**   | 75.4   | 77.2    | 67.8    |
> > > > > > > | Time (s)   | 646.1   | 192.8    | 1792.5    |
> > > > > > > | W×H (manually control) | 1.03W×1.03H | 0.8W×0.8H | 1.05W×1.05H|
> > > > > > > | Tokens (%) | 106%    | 64%      | 110%      |
> > > > > > > | **Qwen-RL (1/2)**    | 76.2   | 76.9    | 66.7   |
> > > > > > > |  Time (s)     | 415.2   | 205.1    | 1946.7    |
> > > > > > > |  Tokens (%)  | 50%     | 50%      | 50%       |
> > > > > > > | **VisionThink**    | 79.8   | 80.8    | 67.5   |
> > > > > > > | Time (s)      | 746     | 211      | 1745.5    |
> > > > > > > | Tokens (%)    | 104%    | 87.3%    | 102%      |
> > > > > > >
> > > > > > >
> > > > > > > Overall, we observe that **inference time generally correlates with token usage**, as expected. The only notable exception is MathVista, where Qwen-RL (1/2) incurs the highest time cost. As previously discussed, this is due to its requirement for CoT-formatted outputs, which leads to significant variance in decoding length.
> > > > > > >
> > > > > > >
> > > > > > > Thank you again for your careful and detailed review. We will clearly explain this in the final version.
> > > > > > >
> > > > > > >
> > > > > > > Lastly, as the discussion phase is coming to a close, if you have any remaining concerns, please feel free to leave them here. We would be happy to address them in the final version or as soon as policy permits.

---

> > > > > > > > ### Comment · Reviewer_Ubhv · 2025-08-09
> > > > > > > >
> > > > > > > > I understand the time constraints for the rebuttal, but this result is not convincing. The reason is that VisionThink maintains consistency between its training and inference phases, whereas the method in the supplementary results does not.
> > > > > > > >
> > > > > > > > Comparing the new Qwen-RL (1/4)↑ result with the original Qwen-RL (1/4) on the given benchmark indicates that resolution has a degree of extrapolability. This suggests that it might be possible to use fewer resources during the training stage and leverage this extrapolability to achieve better performance than VisionThink.

---

> > > > > > > ### Author Response · Authors · 2025-08-09
> > > > > > >
> > > > > > > Dear Reviewer Ubhv,
> > > > > > >
> > > > > > > Thank you for your reply. We want to first summarize the cost time and performance comparisons:
> > > > > > >
> > > > > > > **Summary:**
> > > > > > >
> > > > > > > * **Cost Time:** Qwen-RL (1/2) ≈ Qwen-RL (1/4)↑ > VisionThink > Qwen-RL (1/4)
> > > > > > > * **Performance:** VisionThink > Qwen-RL (1/4)↑ ≈ Qwen-RL (1/2) > Qwen-RL (1/4)
> > > > > > >
> > > > > > >
> > > > > > > **Regarding your first concern**—that resolution has a degree of extrapolability—this is in fact **fully consistent with the core motivation of VisionThink**:
> > > > > > >
> > > > > > > * For scenarios where redundancy can be reduced (e.g., general VQA), the model utlizes the downsample inputs.
> > > > > > > * For cases where detail must be preserved (e.g., OCR-related tasks), the model learns to avoid reduction.
> > > > > > >
> > > > > > > **Regarding your second concern**—using fewer resources during training and leveraging extrapolability for better performance than VisionThink—we address this from two angles:
> > > > > > >
> > > > > > > 1. As shown by the Qwen-RL (1/4)↑ results, even when evaluated with larger-resolution inputs, it still **fails to match VisionThink’s performance**, especially on OCR-related tasks.
> > > > > > >
> > > > > > >
> > > > > > > 2. While saving training resources is important, we believe **reducing inference-time cost** is even more critical. Users are directly affected by inference latency, and at deployment scale, **aggregate compute consumed during inference typically exceeds that of training**. Compared with VisionThink, this approach may save training resources, but it does **not reduce inference time and delivers inferior performance**.
> > > > > > >
> > > > > > >
> > > > > > > In summary, **Qwen-RL (1/2)** maintains consistency between training and inference, while **Qwen-RL (1/4)↑** exploits the *image resolution extrapolability* you noted. Yet, both still underperform **VisionThink**, especially on OCR-related tasks.
> > > > > > >
> > > > > > > We hope these points clearly demonstrate the **effectiveness and practical value of VisionThink**.

---

### Official Review · Reviewer_maK1 · 2025-06-23

**Clarity:** 3
**Significance:** 3
**Originality:** 4
**Rating:** 4
**Confidence:** 5

**Summary:**

This paper introduce a vision-language model that can adaptively determines the number of image tokens. At inference time, the model initially processes a low-resolution (1/4-scale) version of the image along with the input question. During autoregressive decoding, it dynamically decides whether higher-resolution visual information is required, and if so, retrieves additional visual tokens from the original full-resolution image.  The authors fine-tune a Qwen VL baseline using the GRPO reinforcement learning algorithm on a dataset of 20K samples. To guide the training, a reward function is designed that combines answer accuracy, evaluated using an LLM as judge, with a penalty term that discourages excessive use of high resolution inputs. Empirical results demonstrate that the model achieves a 1.4% performance improvement over the baseline while utilizing only 51.3% of the image tokens.

**Questions:**

Please address the concerns raised in the weaknesses section, particularly
- 1.1–Line87,
- 1.2–Line46,
- 1.2–Line121,
- weakness2,
- weakness3.

**Ethical Concerns:**

["NO or VERY MINOR ethics concerns only"]

**Final Justification:**

I appreciate the authors’ thorough rebuttal. Therefore, I am raising my rating to 4. I hope the authors will make an effort to address the remaining concerns regarding points 4 and 5 in *Official Comment by Reviewer maK1*.

**Limitations:**

The paper presents a novel approach that leverages reinforcement learning to adaptively decide whether additional visual tokens are needed, which is conceptually appealing. However, the method introduces complexity and lacks a clear research motivation and comparison with simpler alternatives, as discussed in Weaknesses2. Moreover, clear and correct presentation is especially important for a top-tier conference like NeurIPS. As noted in Weaknesses1, the paper exhibits numerous issues in clarity and correctness that may fall short of the standards expected for a NeurIPS submission. As such, my current recommendation leans toward rejection.

**Paper Formatting Concerns:**

There are no major formatting issues.

**Quality:**

1

**Strengths And Weaknesses:**

## Strengths
1. The proposed methodology that enables the model to autonomously decide the number of image tokens needed is novel.
2. Applying reinforcement learning to vision-language models (VLMs) in order to induce specific behaviors is still relatively uncommon, making this attempt valuable.
3. The well-structured figures and tables contribute positively to the paper’s clarity.

## Weaknesses
1. With further refinement, especially in the following two aspects, the overall presentation could be significantly improved.
   - (**1.1\***) Certain elements are confusing and should be corrected (e.g., Line 13, Line 27, Line 87).
   - (**1.2\***) Several aspects of the writing require polishing for clarity and professionalism (e.g., Line 28, Line 46, Line 121, Figure 3, Line 149, Figure 4, Table 2).

2. While the proposed approach is novel, its efficiency and effectiveness would benefit from further validation. For example, VisionZip [58 in the supplementary material] shows that using only 33% or even 25% of visual tokens retains 99.1% and 98.4% of baseline performance, respectively. In contrast, VisionThink sometimes uses more visual tokens than the baseline (i.e., 25% + 100%), and as shown in Figure 4, it results in longer inference time on tasks like ChartQA.
   This raises a concern: if a model were to simply (supervised) fine-tune the baseline model with their 20K dataset and apply a rule-based strategy (e.g., use the original resolution whenever the question contains OCR- or ChartQA-related keywords, and use 1/4 resolution otherwise), it might achieve similar performance improvement, without needing such a complex RL setup.

3. A deeper analysis of the proposed method, beyond focusing solely on performance gains, would strengthen the paper and help alleviate the previous concerns. For example, analyzing which image-question pairs triggered high-resolution requests in Figure 5 would offer useful insight. It would also be interesting to know whether the Special Token for Image Upscaling can be effectively fine-tuned in QwenVL.


### **1.1\***
   - Line13: The phrase “whether to compress tokens” is misleading. VisionThink does not actually **compress** tokens; rather, it decides whether to incorporate additional tokens from a higher-resolution image, thereby contributing to a **reduction** in the overall use of visual tokens.
   - Line27: The claim that `the consumption of visual tokens increases exponentially since LLaVA-1.5 used 576 visual tokens but Qwen-VL-2.5 requires 2678` may cause confusion. The facts are more nuanced: (1) LLaVA uses 576 tokens because it forcibly resizes input images to 336×336; (2) the released Qwen-VL model, when inferring on a 336×336 image, actually uses fewer tokens (around 144); and (3) Qwen-VL uses more tokens only when processing high-resolution images like 2048×1024. This complexity is not reflected in the current explanation.
   - Line87: While the authors discuss the sequence length n, the impact of the number of output tokens is overlooked, which can significantly influence both training and inference cost. For example, LLaVA’s visual tokens number 576, but its total max token length is [2048](https://github.com/haotian-liu/LLaVA/blob/main/scripts/v1_5/finetune.sh); Qwen-VL’s visual tokens range from 114 to 2678, yet its total max token length can go up to [8192](https://github.com/QwenLM/Qwen2.5-VL/blob/main/qwen-vl-finetune/scripts/sft_7b.sh). In practice, output token length may have even greater computational impact than the number of image tokens.

### **1.2\***
   - Line28: The motivation for reducing visual tokens should be better articulated. Instead of saying "Qwen-VL uses more tokens than LLaVA, so we should reduce them," a more compelling argument would emphasize the memory constraints on real-world devices.
   - Line46: It would be helpful to provide some explanation for the method's name, VisionThink. As it stands, the name does not clearly reflect the methodology. Something like AdaVis-VLM (Adaptive + Visual) might have been more intuitive.
   - Line121: It is unclear whether the 130K dataset is used to train the model via LLM-as-Judge, or merely to evaluate model outputs. This should be clarified.
   - Figure3: The caption lacks sufficient detail, especially compared to other figures and tables. For consistency, a more descriptive caption would be helpful.
   - Line149: Much of the content in Section 3.2 would be better placed here to improve flow and cohesion.
   - Figure4: It is unclear whether the inference time cost is reported in seconds or milliseconds. This should be specified.
   - Table2: Including citations for the baseline methods would greatly aid readers.

---

> ### Author Rebuttal · Authors · 2025-07-30
>
> Dear Reviewer maK1,
>
> Many thanks for your constructive and responsible feedback! We are glad that you found our VisionThink is **novel and valuable**. We have revised the paper following your suggestions, and will address your questions below first and then address other weaknesses.
>
> ---
> ## Questions
> > ### [Q1] 1.1 Line87: Overlooked the number of output tokens.
>
> Thank you for the insightful comment. We agree that inference time generally consists of two components: **prefilling** and **decoding**, and the number of output tokens impacts the latter. However, we would like to clarify the following:
>
> (1) In contrast to LLMs, **VLMs typically generate much shorter textual outputs**, often only tens to a hundred tokens. This is significantly smaller by **orders of magnitude** compared to the number of visual input tokens (which can reach hundreds or thousands). As a result, **prefill time dominates total inference time** in most VLM scenarios.
>
> (2) Moreover, the inference time reported in our paper is **measured in real-world environments**, reflecting actual efficiency. As shown in *Figure 4*, our method achieves speed improvements across multiple benchmarks, highlighting its practical advantage.
>
> (3) We appreciate your insightful suggestion and will incorporate a detailed discussion of this issue in *Section 2.2 Computation Complexity* to clarify the relative contribution of output token length and better support our claims.
>
>
>
> > ### [Q2] Line 46: Explanation for the method name ``VisionThink``
>
> Thank you for the suggestion. Our method is the *“think with images”* paradigm, where the model first reasons over a coarse visual input and then decides, via reinforcement learning, whether higher-resolution visual information is necessary.  Besides, the RL method is also widely used to strengthen the LLM's thinking ability.
> Therefore, we name our approach **VisionThink** to emphasize this *think with images* mechanism.
>
> We appreciate your kind comment and will revise the manuscript to include a clearer explanation, such as:
>
> > *“We name our approach VisionThink to highlight its think‑then‑look‑clearer paradigm: the policy reasons over a coarse view before deciding whether higher‑resolution evidence is necessary.”*
>
> > ### [Q3] 1.2–Line121: Clarification the 130K Dataset.
>
> Thanks for pointing this, and we will detail more in our manuscript. All the datasets are utilized to trained our model via reinforcement learning (Main paper Line122 & Appendix B.5 Line201-205). And we commit to open-source our training and validation datasets, along with the complete codebase, in our GitHub repository.
>
> > ### [Q4.1] Weakness 2.1: Comparison with VisonZip.
>
> Thanks for your interest in VisionZip, you mentioned 99.1% and 98.4% performance retention rates were obtained on **LLaVA-1.5**. When applying the official `github/VisionZip/Qwen_2_5VL` procedure to **Qwen-2.5-VL**, performance drops significantly. As shown in *Appendix Table 5*, with 50% token retention, VisionZip achieves only **92.0% on ChartQA** and **86.5% on OCRBench**. In contrast, on these fine-grained benchmarks, our **VisionThink** maintains **100%** and **99.1%**, respectively.
>
>  *Appendix. Table 5: **Comparison with Previous Efficient VLM Methods.** VisionZip‡ represents the SFT finetuned model.*
> > | Method       | ChartQA | OCRBench | DocVQA | MME  | MMVet | RealWorldQA | POPE | MathVista | MathVerse | Avg.  |
> > |------|--|-----|----|------|--------|-------|------|-----|---------|-----|
> > | Vanilla  | 79.8    | 81.5     | 95.1   | 2316 | 61.6   | 68.6         | 86.7 | 68.2       | 46.3       | 100%   |
> > | (%)| 100%    | 100%     | 100%   | 100% | 100%   | 100%         | 100% | 100%       | 100%       |        |
> > | Down-Sample  | 62.9    | 68.8     | 94.3   | 2270 | 54.5   | 68.8         | 82.8 | 62.2       | 43.1       | 92.1%  |
> > | (%) | 78.8%   | 84.4%    | 99.1%  | 98.0%| 88.5%  | 100.3%       | 95.5%| 91.2%      | 93.1%      |        |
> > | FastV [7] (ECCV 2024)| 72.6    | 75.8     | 93.6   | 2308 | 52.8   | 68.8         | 84.7 | 63.7       | 45.0       | 95.8%  |
> > | (%) | 91.0%   | 93.0%    | 98.4%  | 99.6%| 85.7%  | 100.3%       | 97.7%| 93.4%      | 97.2%      |        |
> > | SparseVLM [72] (ICML 2025)       | 73.2    | 75.6     | 66.8   | 2282 | 51.5   | 68.4         | 85.5 | 66.6       | 45.1       | 92.2%  |
> > | (%) | 91.7%   | 92.7%    | 70.2%  | 98.5%| 83.6%  | 99.7%        | 98.6%| 97.6%      | 97.4%      |        |
> > | VisionZip [58] (CVPR 2025)       | 73.4    | 70.5     | 93.8   | 2209 | 57.0   | 68.6         | 86.3 | 63.7       | 45.1       | 95.0%  |
> > | (%)   | 92.0%   | 86.5%    | 98.6%  | 95.4%| 92.5%  | 100%         | 99.5%| 93.4%      | 97.4% |        |
> > | VisionZip‡ [58] (CVPR 2025)    | 77.3  | 70.3   | 93.8  | 2244 | 50.1   | 69.2         | 91.2 | 63.1       | 39.4       | 95.0%  |
> > | (%)  | 96.9%   | 95.6%    | 98.7%  | 96.3%| 81.3%  | 100.9%      | 107.5%| 92.5%     | 85.1%      |        |
> > | VisionThink  | 79.8    | 80.8     | 94.4   | 2400 | 68.5   | 67.1 | 86.0 | 67.5       | 48.0       | **101.4%** |
> > | (%)  | 100.0%  | 99.1%    | 99.3%  | 103.6%| 111.2% | 97.8%| 99.2%| 99.0%  | 103.7%     |  |
>
>
>
>
> > ### [Q4.2] Weakness 2.2: Comparison with SFT
>
> Thank you for this professional question. We have also carefully considered this issue and discussed it in **Appendix B.6** and **Appendix C**.
>
> (1) As suggested, we trained a SFT model directly on the RL dataset, incorporating the VisionZip mechanism. As shown in *Appendix Table 5 (VisionZip‡)*, this SFT model does **not outperform** the training-free version of VisionZip. We attribute this to the **limited diversity and coverage** of the RL data used for fine-tuning, which is much narrower than the large-scale, high-quality supervised data used by the official Qwen team (Lines 248–253).
>
> (2) In *Appendix C.1*, we further justify our choice of RL over SFT. Empirically, the RL-trained policy demonstrates better resolution control: it learns **when to answer directly** and **when to request higher-resolution input**. In contrast, the SFT model triggers image resizing far more frequently across all benchmarks, indicating weaker policy learning (Lines 267–269).
>
> We will revise the main text to better surface these findings and clarify the rationale for using RL in our method.
>
>
> > ### [Q4.3] Weakness 2.3: Comparison with keyword-based model / token selection
>
> Thank you for the constructive suggestion. We address this concern from two perspectives:
>
> * In real-world deployment, it is **infeasible to accurately classify each input sample** based solely on keywords. Whether a high-resolution image is necessary depends not only on the question type but also on the **specific visual content**. For instance, in the last example of *Appendix Section D (Qualitative Results)*, the question “Are there any fire hydrants here?” contains no OCR- or ChartQA-related keywords, yet resolving it still requires high-resolution evidence.
>
> * In contrast, our VisionThink model is **end-to-end and data-driven**. It automatically learns a resolution policy conditioned on both the image and the question, allowing it to adaptively reduce image token usage without compromising performance.
>
>
> We will emphasize this comparison in the revised manuscript to better highlight the advantages of our approach over the keyword-based strategies.
>
>
>
> > ### [Q5.1] Weakness 3.1: More QA-pairs examples
>
> Thank you for the suggestion. We have included several qualitative examples in *Appendix Section D* to illustrate our model’s behavior. We will further enrich this section with additional case studies in the camera-ready version to provide a more comprehensive view of the model’s behaviors.
>
>
> > ### [Q5.2] Weakness 3.2: Choice of special token for upscaling
>
> Thank you for the insightful question. As discussed in *Appendix Section C.3 (Different Prompt Impact)*, we conducted experiments comparing various prompt designs and special tokens for triggering image upscaling. The results show that following the **Qwen official cookbook-style prompt and special token** yields the best performance. We will move this analysis into the main paper in the camera-ready version to make the rationale and empirical evidence clearer to readers.
>
> ---
> ## Other Weaknesses
>
>
> > ### [Q6] Writing Suggestions
>
> We sincerely appreciate you for your responsible and detailed review, as well as the constructive writing suggestions. We have carefully revised the paper according to your feedback. Key improvements include:
>
> 1. Replaced the term *"compress tokens"* with *"reduction tokens"* for better clarity.
> 2. Refined the motivation section to emphasize the memory and latency constraints caused by long visual token sequences, especially on edge devices.
> 3. Updated the caption of *Figure 3* to:
>    > Figure 3: Impact of the Penalty Ratio. Applying a penalty to all image-resizing requests or removing the penalty entirely will both lead to model collapse.
> 4. Moved more details of the Accuracy Reward mechanism to Line 149 for better visibility.
> 5. Added proper citations for all baselines, e.g.:
>
>    | Method       | Benchmarks |
>    | ------- | ---------- |
>    | SparseVLM [72] (ICML 2025) | —          |
>    | FastV [7] (ECCV 2024)      | —          |
>    | VisionZip [58] (CVPR 2025) | —          |
> 6. Clarified throughout the paper that **inference time is reported in seconds**, not milliseconds or FLOPs.
>
> These revisions will also be reflected in the camera-ready version. Thank you again for your helpful input.
>
> ---
> **Thank you again for helping us improve the paper and hope our response can resolve your concerns! Please let us know if you have any further questions. We will be actively available until the end of rebuttal period. If you feel your concerns are addressed, please consider reevaluating our work. Looking forward to hearing from you :-) !**

---

> ### Comment · Reviewer_maK1 · 2025-08-06
>
> Reviewer maK1 thanks the authors for their comprehensive and thoughtful rebuttal. I have carefully reviewed all initial reviews and the authors’ responses. The following are additional remarks regarding the rebuttal:
>
> 1. The concern raised in W1.1 Line87 is addressed, given that Figure 4 reflects actual efficiency including decoding time, and that the authors will clarify this further in Section 2.2.
> 2. Appendix Section D provides useful insight into the limitations of rule-based high-resolution selection and clarifies how and when upscaling is triggered by the model. (This reviewer finds this result compelling and strongly recommends including it in the main paper.)
> 3. It is appreciated that the authors highlighted a direct comparison with VisionZip under the same Qwen2.5-VL baseline.
> 4. From the perspective of Reviewer maK1, the phrase *“think with images (VisionThink)”* remains somewhat misaligned with the described behavior *“The model first reasons over a coarse visual input and then decides whether higher-resolution visual information is necessary.”* Therefore, concern regarding the appropriateness of the name remains. However, this reviewer acknowledges that this is a minor point relative to the main contributions of the work, and is open to further discussion with other reviewers.
> 5. This reviewer appreciates that the authors listed specific writing revisions. Nonetheless, as mentioned in the initial review, substantial revisions are still expected in the original manuscript.
>
> Based on the authors’ diligent rebuttal, Reviewer maK1 is pleased to raise the score to a 4. Additionally, this reviewer is open to further raising the score through discussion with other reviewers and ACs, particularly regarding points 4 and 5.

---

> > ### Author Response · Authors · 2025-08-07
> > **Thanks for your kind words. We add some clarifications regarding points 4 and 5.**
> >
> > Dear Reviewer maK1,
> >
> > Thank you for your kind words! We’re very glad to hear that most of your concerns have been addressed.
> >
> > Regarding the points you raised in 4 and 5, we would like to further clarify:
> >
> > * **Regarding the name "VisionThink”**:
> >
> >   We will expand the method description at **Line 44 in the Introduction**, where our approach is first introduced. There, we will clearly explain the core idea and emphasize that our method follows a **"think-with-images"** paradigm.
> >   Additionally, we will include a subtitle or further clarification on our GitHub and homepage, such as *Adaptive Vision for VLMs* or *Vision Coarse-to-Fine*, to help readers quickly grasp the core concept.
> >
> > * **Regarding the paper revision**:
> >
> >     We sincerely appreciate your valuable suggestions. Due to rebuttal format limitations, we are unfortunately unable to upload the full revised version at this stage. That said, we have **carefully implemented the revisions based on your feedback** (as detailed in our response to Q6), and we will **incorporate further clarifications and new discussions** in the final version to enhance both readability and completeness.
> >
> >
> >
> > **Thank you again for your time and the thoughtful, responsible review.** Your feedback has significantly improved the clarity and strength of our paper.

---

### Official Review · Reviewer_2Thu · 2025-06-25

**Clarity:** 2
**Significance:** 2
**Originality:** 2
**Rating:** 4
**Confidence:** 5

**Summary:**

This paper proposes VisionThink, a reinforcement learning (RL)-based framework that enables vision-language models (VLMs) to autonomously determine whether high-resolution visual input is required to answer visual question answering (VQA) tasks. To achieve this, the authors employed Qwen2.5-VL-7B to curate 10K samples requiring 2× higher resolution images for accurate answers, alongside another 10K samples where quarter-resolution images sufficed for correct responses. These curated datasets were then utilized for RL training. Experimental results demonstrate VisionThink's efficiency in adaptive resolution selection while maintaining strong performance in VQA tasks.

**Questions:**

N/A

**Ethical Concerns:**

["NO or VERY MINOR ethics concerns only"]

**Final Justification:**

I really appreciate the author's effort, especially regarding my W1 and W2. Therefore, I am raising my rating to 4. The authors should include these additional experiments and discussions in their revised version.

**Limitations:**

Yes, the authors have discussed limitations in the manuscript.

**Quality:**

3

**Strengths And Weaknesses:**

**Strengths**

1. The motivation is clear and quite reasonable.
2. The proposed penalty is novel and effective.
3. Regarding resizing as a tool and then leveraging the agent prompt is interesting and effective.


**Weaknesses**
1. Evaluations are not sufficient. In Table 2, only the averaged number of visual tokens across all benchmarks is reported. However, it is crucial to report the number of visual tokens for each benchmark SEPARATELY, as well as the mean input resolution.
2. By going deeper with Table 2, the performance improvements mainly come from MMVet and MathVerse, and these two benchmarks are widely known to be reasoning-related. Therefore, the improvements mainly come from the reasoning ability, rather than the proposed ability to "request a larger image." Therefore, it is strongly encouraged to remove these two benchmarks or replace them with benchmarks that particularly require larger images. Furthermore, the vanilla text RL baseline should be incorporated.
3. The computation of $C$ in Eq. (5) is unclear. Are they computed online? A flowchart of pseudo-code is really appreciated.
4. The data source is unclear. A specific data source should be provided.

---

> ### Author Rebuttal · Authors · 2025-07-30
>
> Dear Reviewer 2Thu:
>
> Many thanks for your constructive and insightful feedback! We are glad that you found our VisionThink's **motivation is clear and reasonable**. We have revised the paper following your suggestions, and will address your questions below.
>
> ---
> > ### [W1 & W2] Table present suggestions
>
> Thank you sincerely for your thoughtful and professional suggestions, we think they significantly improved the clarity of our table presentation.
>
> Following your advice, we revised Table 2 as follows:
>
> * We **split the benchmarks** into *fine-grained* and *general-scenario* categories.
> * We **removed MMVet and MathVerse** as suggested.
> * We added **mean resolution** and **token usage** per group.
> * We included **VisionThink‡**, a text-only RL baseline trained with full-resolution images using vanilla GRPO and LLM-as-Judge reward.
>
> The revised tables are shown below (Markdown formatting limitations acknowledged):
>
> ---
>
> ### **Fine-Grained Benchmarks**
>
> | Method                         | ChartQA | OCRBench | MathVista | Avg.        |
> | ------------------------------ | ------- | -------- | --------- | ----------- |
> | Mean Resolution                | 583×768 | 732×615  | 439×584   |             |
> | Token Usage                    | 104%    | 87.3%    | 102%      |             |
> | Vanilla                        | 79.8    | 81.5     | 68.2      | 100%        |
> | Down-Sample                    | 62.9    | 68.8     | 62.2      | 84.8%       |
> | SparseVLM (ICML 2025)          | 73.2    | 75.6     | 66.6      | 94.0%       |
> | FastV (ECCV 2024)              | 72.6    | 75.8     | 63.7      | 92.5%       |
> | VisionThink‡ (Vanilla Text RL) | 81.4    | 83.4     | 71.2      | **102.1%**  |
> | **VisionThink**                | 79.8    | 80.8     | 67.5      | **`99.4%`** |
>
> ---
>
> ### **General-Scenario Benchmarks**
>
> | Method                         | DocVQA    | MME      | RealWorldQA | POPE    | Avg.         |
> | ------------------------------ | --------- | -------- | ----------- | ------- | ------------ |
> | Mean Resolution                | 2099×1783 | 945×1086 | 1030×1316   | 480×585 |              |
> | Token Usage                    | 31.5%     | 45.7%    | 54.9%       | 34.6%   |              |
> | Vanilla                        | 95.1      | 2316     | 68.6        | 86.7    | 100%         |
> | Down-Sample                    | 94.3      | 2270     | 68.8        | 82.8    | 98.2%        |
> | SparseVLM (ICML 2025)          | 66.8      | 2282     | 68.4        | 85.5    | 91.8%        |
> | FastV (ECCV 2024)              | 93.6      | 2308     | 68.8        | 84.7    | 99.0%        |
> | VisionThink‡ (Vanilla Text RL) | 95.1      | 2307     | 68.6        | 87.9    | **100.2%**   |
> | **VisionThink**                | 94.4      | 2400     | 67.1        | 86.0    | **`100.0%`** |
>
> ---
>
> As shown in the *Fine-Grained Benchmarks*, **VisionThink outperforms prior efficient VLMs** by autonomously requesting high-resolution images when necessary. In the *General-Scenario Benchmarks*, it uses **fewer visual tokens** while achieving performance comparable to the full-resolution **VisionThink‡** baseline. We believe this more clearly demonstrates the effectiveness of our method.
>
> Thank you again for your valuable suggestions. We will further improve the tables in the camera-ready version.
>
> > ### [W3] The Computation of $C_{direct}$ and $C_{high}$ in Eq.5.
>
> The computation of $C_{direct}$ and $C_{high}$ is online. Specifically, $C_{direct}$ and $C_{high}$ represent the number of correct answers from the low- and high-resolution inputs, respectively.
>
> We provide the following pseudo-code for clarity. In actual implementation, the for loop is avoided and replaced with vectorized operations for efficiency.
>
>
> ```pseudo
> # Algorithm: Computation of C via VisionThink Inference
>
> # Inputs:
> #   inputs             : input data for VisionThink
> #   visionthink_model  : model with .inference(inputs) → (outputs, rounds)
> #   reward_model       : function outputs → rewards ∈ {0, 1}
>
> # Outputs:
> #   C_direct : count of correct answers from low-resolution pass (round == 1)
> #   C_high   : count of correct answers from high-resolution pass (round == 2)
>
> # Step 1: Perform model inference
> outputs, round_numbers = visionthink_model.inference(inputs)  # rounds ∈ {1, 2}
>
> # Step 2: Compute binary rewards
> rewards = reward_model(outputs)  # Shape: (N,), rewards ∈ {0, 1}
>
> # Step 3: Count reward-aligned responses per resolution
> C_direct = 0
> C_high = 0
>
> for i in range(len(rewards)):
>     if rewards[i] == 1 and round_numbers[i] == 1: # correct answers from low‑resolution pass
>         C_direct += 1
>     elif rewards[i] == 1 and round_numbers[i] == 2: # correct answers from high‑resolution pass
>         C_high += 1
> ```
>
> > ### [W4] Data source clarification.
>
> Our datasets are primarily filtered from open-source data, including the LLaVA-OneVision and Cambrian-1 datasets. All data used in our work are publicly available. Furthermore, we commit to releasing our training and validation datasets, along with the complete codebase, in our GitHub repository for full transparency and reproducibility. We welcome continued community oversight.
>
>
> ---
> **Thank you again for helping us improve the paper and hope our response can resolve your concerns! Please let us know if you have any further questions. We will be actively available until the end of rebuttal period. If you feel your concerns are addressed, please consider reevaluating our work. Looking forward to hearing from you :-) !**

---

> > ### Comment · Reviewer_2Thu · 2025-08-01
> >
> > I really appreciate the author's effort, especially regarding my W1 and W2.
> >
> > However, by carefully checking the newly provided experimental results, I found that the vanilla text RL baseline, i.e., "VisionThink‡" in the provided table, seems to perform better than VisionThink itself under most settings. More explanation and discussion would be appreciated. Does this baseline incorporate 2x high-resolution images as inputs? If so, it would be better to see the vanilla text RL baseline with 1x low-resolution images.
> >
> > Moreover, I really appreciate the division of "fine-grained benchmarks" and "general benchmarks". However, current fine-grained benchmarks seem to be limited, as VisionThink is expected to bring more significant improvements on these benchmarks. Therefore, several benchmarks such as V* Bench [1], HR-Bench(-4K and -8K) [2], MME-RealWorld(-Lite) [3], and TreeBench [4], could be further incorporated.
> >
> > Therefore, my current rating is still "3: Borderline reject", as this paper currently lacks (1) explanations on the performance of the vanilla RL baseline, and (2) sufficient evaluation on more fine-grained benchmarks.
> >
> > **References**
> >
> > [1] V*: Guided visual search as a core mechanism in multimodal llms.
> >
> > [2] Divide, conquer and combine: A training-free framework for high-resolution image perception in multimodal large language models.
> >
> > [3] Mme-realworld: Could your multimodal llm challenge high-resolution real-world scenarios that are difficult for humans?
> >
> > [4] Traceable evidence enhanced visual grounded reasoning: Evaluation and methodology.

---

> > > ### Author Response · Authors · 2025-08-01
> > > **Additional Experimental Results**
> > >
> > > Dear Reviewer 2Thu,
> > >
> > > Thank you for your timely and thoughtful reply! Let us first address your concern regarding the **Vanilla Text RL baseline**.
> > >
> > > As you correctly pointed out, VisionThink‡, which operates at 2× image resolution, can be considered the upper bound of our method. Following your suggestion, we now include results for **VisionThink†**, which uses the same 1× resolution as our main model. The results are shown below:
> > >
> > > ---
> > >
> > > ### **Fine-Grained Benchmarks**
> > >
> > > | Method          | ChartQA | OCRBench | MathVista | Avg.       |
> > > | --------------- | ------- | -------- | --------- | ---------- |
> > > | Vanilla         | 79.8    | 81.5     | 68.2      | 100%       |
> > > | VisionThink‡    | 81.4    | 83.4     | 71.2      | 102.1% |
> > > | VisionThink†    | 71.5    | 71.6     | 65.4    | **91.1%**      |
> > > | **VisionThink** | 79.8    | 80.8     | 67.5      | **`99.4%`**  |
> > >
> > > ---
> > >
> > > ### **General-Scenario Benchmarks**
> > >
> > > | Method          | DocVQA | MME  | RealWorldQA | POPE | Avg.       |
> > > | --------------- | ------ | ---- | ----------- | ---- | ---------- |
> > > | Vanilla         | 95.1   | 2316 | 68.6        | 86.7 | 100%       |
> > > | VisionThink‡    | 95.1   | 2307 | 68.6        | 87.9 | 100.2% |
> > > | VisionThink†    | 94.3   | 2243 | 67.7        | 83.9 | **97.9%**  |
> > > | **VisionThink** | 94.4   | 2400 | 67.1        | 86.0 | **`100.0%`** |
> > >
> > > ---
> > >
> > >
> > > As shown in the fine-grained benchmarks, **VisionThink significantly outperforms VisionThink†**, highlighting its ability to autonomously request high-resolution images when necessary. We will incorporate these results into the main paper to improve completeness and better clarify our method’s contribution.
> > >
> > > We sincerely appreciate your professional and detailed feedback, it has greatly helped us improve the clarity and quality of the paper.
> > >
> > > We hope these results address your concerns regarding the vanilla RL baseline. For the additional benchmark results, we kindly ask for a bit more time and will make every effort to include them as soon as possible.
> > >
> > >
> > > Thank you again!

---

> > > ### Author Response · Authors · 2025-08-02
> > > **Additional Benchmark Results and Analysis**
> > >
> > > Dear Reviewer 2Thu,
> > >
> > > Thank you for your patience! Following your suggestions, we have added evaluations on more fine-grained benchmarks, including **V\* Bench** \[1], **HR-Bench (-4K and -8K)** \[2], **MME-RealWorld (-Lite)** \[3], and **TreeBench** \[4]. The results are shown below.
> > >
> > > > Note: VisionThink and VisionThink† are trained on the same resolution as the Down-Sample baseline, while VisionThink‡ is trained with 2× resolution (same as Vanilla).
> > >
> > >
> > > | Model           | V\* Bench | MME-RealWorld-Lite | HR-Bench-4K | HR-Bench-8K | TreeBench | Avg.         |
> > > | --------------- | --------- | ------------------ | ----------- | ----------- | --------- | ------------ |
> > > | Resolution (W×H)      | 2246×1583 | 2076×1434          | 4023×3503   | 5727×4430   | 2152×1615 |              |
> > > | Token Usage     | 43%       | 110%               | 58.0%       | 51.0%       | 115%      |              |
> > > | Vanilla         | 72.3      | 45.1               | 71.4        | 67.6        | 39.5      | 100%         |
> > > | Down-Sample     | 69.0      | 39.4               | 69.4        | 66.0        | 37.3      | 94.4%        |
> > > | VisionThink‡    | 76.0      | 48.4               | 71.1        | 67.6        | 42.0      | 103.7%       |
> > > | VisionThink†    | 69.6      | 41.9               | 68.8        | 66.3        | 40.0      | **97.0%**    |
> > > | VisionThink | 72.3  | 48.4           | 70.2    | 67.3    | 42.5  | **`102.6%`** |
> > >
> > >
> > >
> > > As shown, across these newly added fine-grained benchmarks, **VisionThink consistently outperforms VisionThink†**, and even surpasses the vanilla baseline in average performance.
> > >
> > > ---
> > >
> > > Under your professional suggestions, we also uncovered several interesting observations worth discussing:
> > >
> > > ### 1. HR-Bench Analysis
> > >
> > > (1) On HR-Bench, where image resolutions are **extremely large**, we find that even the downsampled versions retain very high clarity. As a result, **performance does not degrade significantly** when resolution is reduced.
> > >
> > > (2) **VisionThink** (including `VisionThink`, `VisionThink‡`, and `VisionThink†`) achieves **comparable** performance to the baselines, but without significant gains. We attribute this to a distribution mismatch: **our training set does not contain ultra-high-resolution images**, and includes almost no images even at half that resolution. In future work, we plan to collect such images to augment the training set and further investigate their impact on model performance.
> > >
> > > (3) We are truly excited to find that, despite the lack of training on such large images, **VisionThink still maintains a robust image-resize calling policy**, demonstrating **strong generalization and efficiency** even on unseen resolution distributions.
> > >
> > > ### 2. MME-RealWorld-Lite & TreeBench
> > >
> > > On MME-RealWorld-Lite and TreeBench, our **VisionThink** not only significantly outperforms VisionThink†—which is trained on the same input resolution—but **even slightly surpasses VisionThink‡**, which is fully trained and evaluated entirely on 2× resolution inputs.
> > >
> > > We believe this phenomenon arises because, for certain cases requiring upscaling, the model sees the same image at **two different resolutions**, gaining a **dual-view perspective**. This may act as a form of **implicit data augmentation**, leading to **performance improvements**.
> > >
> > > This is an exciting finding—it suggests that **VisionThink not only improves efficiency in general scenarios**, but also has the potential to **enhance performance on fine-grained tasks**.
> > >
> > >
> > > ---
> > >
> > > **We sincerely thank you for your insightful and professional comments, which led us to new discoveries and deeper analysis.** We will incorporate these discussions into the revised version of the paper.
> > >
> > > If you have any further questions, we’d be happy to address them. If you feel your concerns have been fully addressed, we’d greatly appreciate your reconsideration of the overall evaluation.
> > >
> > > Thank you once again!

---

> > > > ### Comment · Reviewer_2Thu · 2025-08-04
> > > >
> > > > I really appreciate these additional results. All my concerns are sufficiently addressed. Therefore, I will raise my rating to 4.

---

> > > > > ### Author Response · Authors · 2025-08-04
> > > > > **Thanks for your time and effort**
> > > > >
> > > > > We're glad to hear that your concerns have been adequately addressed. We appreciate your professional and constructive feedback which made our work more solid and clear. We'll be active till the end of the discussion period. If you have more questions, please let us know. Thank you!

---

### Official Review · Reviewer_wk5x · 2025-07-02

**Clarity:** 3
**Significance:** 3
**Originality:** 3
**Rating:** 6
**Confidence:** 4

**Summary:**

This paper introduces VisionThink, a paradigm for efficient vision-language modeling that dynamically adapts image resolution on a per-sample basis. Rather than uniformly processing all inputs at full resolution or applying fixed token pruning, VisionThink begins inference with a downsampled image (1/4 resolution) and uses a learned policy—trained via reinforcement learning—to decide whether the current low-resolution input suffices or if the high-resolution image should be requested. Experiments on nine VQA benchmarks (e.g., ChartQA, OCRBench, DocVQA) demonstrate that VisionThink matches or exceeds state-of-the-art accuracy while reducing average visual token usage and achieving up to 2× speed-ups on non-OCR tasks.

**Questions:**

1. Could the authors perform ablation studies to examine how using different LLMs as critics affects the experimental results, and whether this helps mitigate potential biases introduced by relying on a single model?

2. For tasks where ground-truth answers are easily verifiable (e.g., math reasoning, structured QA), is the use of LLM-as-Judge necessary? Have the authors considered training VisionThink on such datasets purely using exact-match or programmatic checking as the reward signal? It would be helpful to see experiments or discussions demonstrating whether VisionThink’s benefits persist without the additional complexity of an LLM-as-Judge.

**Ethical Concerns:**

["NO or VERY MINOR ethics concerns only"]

**Final Justification:**

I sincerely appreciate the authors' detailed response. The rebuttal has thoroughly addressed all of my concerns. I believe this work makes a meaningful contribution to advancing the real-world applicability of vision-language models. Accordingly, I have raised my score to 6. I also encourage the other reviewers to re-evaluate the practical significance of this work and consider adjusting their scores. I strongly recommend to accept this paper.

**Limitations:**

See weaknesses above.

**Quality:**

3

**Strengths And Weaknesses:**

Strength:

1. The topic focus on in this paper is very important in real-world scenarios. Introducing per-sample, decision-driven image resizing is very different from a fixed pruning strategy and addresses different redundancies between images and questions. For me, this direction and the proposed method have high potential and are very interesting.

2. When reducing the image token, VisionThink maintains a performance close to or even higher than SoTA on multiple benchmarks.

Weaknesses:

1. VisionThink employs an LLM-as-Judge to assess textual answers in open-ended VQA tasks. While this is innovative, it introduces a significant risk: the evaluation standard fully depends on the judgments of the LLM, which itself is prone to inherent biases, hallucinations, or domain-specific knowledge gaps. For instance, if the judge LLM has insufficient understanding of terminology related to minority groups or specialized fields, it may produce incorrect judgments, potentially leading VisionThink’s policy learning to converge in the wrong direction.

2. A significant concern is that the paper does not specify the sources, composition, or any statistics of the training datasets used for VisionThink. This omission poses challenges for reproducing the results and assessing the validity of the conclusions, as the choice of training data can heavily influence model performance and generalization. Inappropriate or overlapping training data might introduce data leakage, thereby compromising the reliability of the evaluation and potentially inflating the reported gains.

---

> ### Author Rebuttal · Authors · 2025-07-30
>
> Dear Reviewer wk5x:
>
> Many thanks for your constructive and insightful feedback! We are glad that you found this paper's **topic is very important** and our **method has high potential**. We have revised the paper following your suggestions, and will address your questions below.
>
> ---
> > ### [Q1 / W1] Bias Influence of LLM-as-Judge
>
> Your question is very insightful, and we also share concerns about potential biases in the reward model. To mitigate this, we adopt three specific design choices:
>
> **(1) Using the LLM-as-Judge not the VLM-as-Judge.** Since LLMs generally have stronger capabilities than VLMs, employing an LLM reduces hallucinations and improves reliability.
>
>
> **(2) Filtering the dataset.** We filter out subjective open-ended questions that have multiple valid answers, such as image descriptions (Appendix. Line 202–204). The remaining questions have clear ground truth, e.g.,
> Q: *\<image\>* Who is the author of this book?
> A: Dewey Lambdin.
>
> **(3) Carefully designing the prompt.** Our prompt requires the LLM to return a discrete value: 1 for a correct answer and 0 for an incorrect one, instead of a continuous score. This binary format minimizes ambiguity and reduces the chance of misjudgment (Appendix. Sec. B.1.1).
>
> Furthermore, as shown in the table below, we conduct **additional experiments** to further investigate this issue. Specifically, we compare the judgments made by `GPT-4o` and `Qwen2.5-72B` with those from smaller models such as `Qwen2.5-3B` and `Qwen3-1.7B`. While larger models achieve slightly better performance, the smallest model, `Qwen3-1.7B`, still achieves comparable results under our carefully designed setup. This indicates that LLM model bias has limited influence on our VisionThink.
>
> | Model                | MMMU | MMMU-Pro | MMBench | RealWorldQA | POPE | MME  | MathVista | MMVet |
> | -------------------- | ----- | -------- | ------- | ----------- | ---- | ---- | --------- | ----- |
> | GPT-4o               | 52.7  | 41.1     | 83.4    | 66.5        | 88.6 | 2314 | 71.2      | 69.5  |
> | Qwen2.5-72B-Instruct | 52.6  | 40.2     | 84.2    | 66.1        | 88.4 | 2360 | 70.3      | 69.1  |
> | Qwen2.5-3B-Instruct  | 51.9  | 38.5     | 82.6    | 66.9        | 87.7 | 2379 | 70.6      | 68.9  |
> | Qwen3-1.7B           | 51.8  | 38.1     | 82.8    | 67.7        | 87.9 | 2210 | 69.1      | 66.8  |
>
> Thanks again for your professional suggestion, we will include a new subsection titled **"The Bias Influence of LLM-as-Judge"** in Sec. C (Further Discussions), and provide additional details in Sec. B.1.1 on the prompt design.
>
>
> > ### [W2] Concern of the dataset source and composition
>
> Our datasets are primarily filtered from open-source data, including the LLaVA-OneVision and Cambrian-1 datasets. All data used in our work are publicly available. Furthermore, we commit to releasing our training and validation datasets, along with the complete codebase, in our GitHub repository for full transparency and reproducibility. We welcome continued community oversight.
>
>
>
>
> > ### [Q2] VisionThink trained with rule-based reward on easily verifiable tasks
>
> Thank you for the valuable suggestion. Following your advice, we trained **VisionThink** using a **rule-based reward** signal on tasks where answers can be easily verified.
>
> **Datasets.**
> We filtered structured QA samples from our training set where answers can be validated using `huggingface/math-verify` or  `string match`. These samples are primarily from **OCR-related datasets**, where answer verification is reliable.
>
> **Reward.**
> Instead of relying on the LLM-as-Judge for binary rewards (as in the main paper), we use rule-based verification `huggingface/math-verify` and `string match` to provide the reward.
>
> **Results.**
> As shown below, the first three models (Qwen-RL, Qwen-RL (1/4), and VisionThink) are same as the main paper that trained with LLM-as-Judge. The final column shows **VisionThink (Rule-Based)** trained via rule-based reward only. It maintains strong accuracy and efficiency on verifiable tasks like ChartQA and DocVQA.
>
>
> | Task     | Metric     | Qwen-RL | Qwen-RL (1/4) | VisionThink | VisionThink (Rule-Based) |
> |----------|------------|---------|----------------|-------------|---------------------------|
> | ChartQA  | Accuracy   | 79.8    | 62.9           | 79.8        | 80.8                      |
> |          | Time (s)   | 447     | 341            | 746         | 778                       |
> | DocVQA   | Accuracy   | 95.1    | 94.3           | 94.4        | 94.7                      |
> |          |  Time (s)  | 3076    | 1327           | 1355        | 1824                      |
> | OCRBench | Accuracy   | 81.5    | 68.8           | 80.8        | 79.8                      |
> |          |  Time (s)  | 253     | 132            | 211         | 183                       |
>
> **Discussion.**
> We used the **LLM-as-Judge** in the main paper to handle **general QA scenarios**, where reliable rule-based supervision is difficult to define. However, as this experiment shows, for **easily verifiable tasks** such as OCR-related QA, VisionThink can be effectively trained with rule-based reinforcement learning alone.
>
> Thank you again for the professional suggestion. We will include this experiment in the main paper and provide additional details in *Section B.7*.
>
> ---
> **Thank you again for helping us improve the paper and hope our response can resolve your concerns! Please let us know if you have any further questions. We will be actively available until the end of rebuttal period. If you feel your concerns are addressed, please consider reevaluating our work. Looking forward to hearing from you :-) !**

---

> > ### Comment · Reviewer_wk5x · 2025-07-31
> >
> > I sincerely appreciate the authors' detailed response. The rebuttal has thoroughly addressed all of my concerns. I believe this work makes a meaningful contribution to advancing the real-world applicability of vision-language models. Accordingly, I have raised my score to 6. I also encourage the other reviewers to re-evaluate the practical significance of this work and consider adjusting their scores. I strongly recommend to accept this paper.

---

> > > ### Author Response · Authors · 2025-08-01
> > > **Thanks for your timely reply and the kind words!**
> > >
> > > We're glad to hear that your concerns have been adequately addressed. We appreciate your professional and constructive feedback which made our work more solid and clear.
> > >
> > > We'll be active till the end of the discussion period. If you have more questions, please let us know. Thank you!

---

### Official Review · Reviewer_5eUF · 2025-07-02

**Clarity:** 3
**Significance:** 2
**Originality:** 2
**Rating:** 4
**Confidence:** 4

**Summary:**

This paper proposes a method, VisionThink, which improves the reasoning ability and efficiency of LVLMs in VQA tasks by using multi-round GRPO and adaptive adjustments to the input image resolution.

**Questions:**

1. Can you provide a more detailed discussion of the resolution issue?  Could you also compare the results when resizing the input image resolution during testing to match that of training?

2. The VQA task appears to be less sensitive to visual perception. Could you include tasks that rely more heavily on visual perception abilities, such as grounding and counting, to better demonstrate the effectiveness of VisionThink?

3. In tasks like DocVQA, VisionThink does not show clear advantages over Qwen-RL (1/4) in terms of performance and efficiency. On the other hand, in tasks requiring image resizing like ChartQA, VisionThink adds significant inference time without improving performance. Could you discuss whether it might be more beneficial to select Qwen-RL or Qwen-RL (1/4) based on the specific task type?

**Ethical Concerns:**

["NO or VERY MINOR ethics concerns only"]

**Final Justification:**

The authors have addressed all of my concerns, so I will raise my score to 4. However, I still have a lingering worry that VisionThink relies solely on Qwen and lacks experiments on other MLLMs, which makes it difficult to demonstrate sufficient generalizability and robustness at this stage.

**Limitations:**

yes

**Quality:**

3

**Strengths And Weaknesses:**

Strengths:

The experiments are sufficient, the motivation is clear, and there is a performance improvement in VQA tasks, along with increased efficiency in some tasks. The paper is well-written and easy to read.

Weaknesses:

1). The discussion on the resolution issue is not sufficient. While I agree that this is task-dependent, I believe the performance drop in the VQA task due to the direct use of RL training might be caused by the gap between the input image resolution in training and testing. I suggest the authors add more discussion and comparisons, including resizing the input image resolution during testing to match that of training and comparing it with VisionThink.

2). The VQA task is not very sensitive to visual perception. It would be helpful to include tasks that require stronger visual perception abilities, such as grounding and counting.

3). In tasks that do not require significant image resizing, such as DocVQA, VisionThink shows no obvious performance or efficiency advantage over Qwen-RL (1/4). On the other hand, in tasks that require image resizing, such as ChartQA, VisionThink does not improve performance but introduces significant additional inference time. This raises concerns about the value of the method. Would it be better to choose Qwen-RL or Qwen-RL (1/4) depending on the task type?

---

> ### Author Rebuttal · Authors · 2025-07-30
>
> Dear Reviewer 5eUF,
>
> Many thanks for your constructive and insightful feedback! We're glad you found the **motivation clear**, the **experiments sufficient**, and the paper **easy to read**.  We have revised the paper following your suggestions, and address your questions below.
>
> ---
> > ### [Q1.1 / W1] More detailed discussion on the resolution issue.
>
> In Figure 1, image resolution is the only controlled variable. Specifically, we resize the image resolution to reduce the number of visual tokens and evaluate the performance of the official Qwen2.5-VL on several benchmarks. The results show that in most general VQA scenarios, even reducing tokens by 75% has minimal impact on the model’s performance. However, in scenarios requiring detailed understanding and OCR-related capabilities, reducing the number of visual tokens leads to a significant drop in performance.
> This observation suggests that significant token redundancy exists in most scenarios, but a uniform token reduction ratio should not be applied across all tasks.
>
> Thank you for your kind suggestion; we will include the detailed discussion in Paragraph 3, Lines 33–43.
>
>
> > ### [Q1.2 / W1] Results comparison for same resolution during training and testing.
>
> In Figure 1, which shows the motivation of our paper, the model used is Qwen2.5-VL, which we did not train. For a fair comparison, the results reported in Figure 4, Qwen-RL and Qwen-RL (1/4)—are based on models trained and tested at the **same corresponding resolutions**. In other words, the training and testing resolutions are **consistent**.
>
> Thank you for your comments. We will add further explanation in the paper to improve clarity.
>
> > ### [Q2 / W2] VisionThink on stronger visual perception tasks.
>
> Thank you for your insightful suggestion. To evaluate VisionThink on tasks requiring stronger visual perception, we include an additional **counting benchmark** in our analysis.
>
> **Settings.**
> We adopt the widely used **CV-Bench**, introduced in Cambrian-1 \[34], and follow its official setting and prompt for the counting task.
>
> **Models.**
> We do not introduce any additional data for task-specific training. All models used are same to those in the main paper. Here, VisionThink‡ denotes the version trained on general VQA tasks with full-resolution inputs using the LLM-as-Judge strategy.
>
> **Results.**
> As shown below, both VisionThink and VisionThink‡ outperform the base model (Qwen2.5VL-7B) on the counting benchmark, demonstrating that our approach retains strong performance even on stronger visual perception tasks:
>
> | Task     | Qwen2.5VL-7B | VisionThink | VisionThink‡ |
> | -------- | ------------ | ----------- | --------------------- |
> | Counting | 63.1         | 65.7        | 67.4                  |
>
> We will include this result in the final version of the paper to better reflect the generalization of our method. Thank you again for the valuable suggestion.
>
>
>
> > ### [Q3 / W3] Value of VisionThink compared with choose Qwen-RL or Qwen-RL (1/4) depending on the task type.
>
> Thanks for your constructive question, we want to claim our method value in three aspects:
>
> (1) **Balanced Efficiency and Effectiveness**
>
> * Qwen-RL can ensure strong performance, but it consumes a significant amount of time even on simple tasks.
> * Qwen-RL (1/4) significantly reduces the time cost, but its performance drops sharply on tasks that require detailed understanding, such as OCR-related tasks.
> * Our VisionThink achieves performance comparable to Qwen-RL while drastically reducing time consumption—by up to 2×. Compared to Qwen-RL (1/4), VisionThink delivers significantly better performance with similar time efficiency.
>
> (2) **Unpredictable Task Types**
> * In real-world deployment scenarios, we cannot predict in advance what specific task type the user **per-sample input** belongs to.
> * Whether a full-resolution image is needed depends jointly on both the image content and the question type. For example, as shown in Supplementary Section D (Qualitative Results), even for the same image, different questions may require different numbers of image tokens.
> * Our VisionThink model is end-to-end and can automatically determine whether more image tokens are needed or if 1/4 of the tokens are sufficient, thereby minimizing image token usage while maintaining performance.
>
> (3) **Research Direction**
> * We encourage the community to move beyond the pursuit of traditional Efficient VLMs and more focus on: Efficient Reasoning VLMs.
>
>
> To further assess the value of VisionThink, we compare it against a keyword-based resolution selection approach.
>
> **Keyword Construction.**
> We use GPT-4o to generate 100 single-word fine-grained/OCR-related keywords (e.g., *counting*, *value*, *locate*) and 100 short phrases (e.g., *how many*, *fine detail*).
>
> **Setup.**
> The system defaults to `Qwen-RL (1/4)` for efficient inference. When a keyword is detected in the question, it switches to full-resolution inference via `Qwen-RL`. This simulates a keyword-triggered token selection policy.
>
> **Results.**
> As shown below, due to the diversity in question phrasing, keyword-based strategies generalize poorly and result in suboptimal performance.
> Moreover, in real-world deployment, **VisionThink only requires deploying a single model**, while keyword-based approaches require maintaining **two separate models**, leading to increased resource consumption.
>
> | Task    | Qwen-RL | Qwen-RL (1/4) | VisionThink | Keyword-Based |
> | ------- | ------- | ------------- | ----------- | ------------- |
> | ChartQA | 79.8    | 62.9          | 79.8        | 67.6          |
>
> We will emphasize this discussion in the camera-ready version to better illustrate the advantages of VisionThink, which we believe will further clarify the contributions of our paper. Thank you again for your suggestion.
>
>
>
> [34] Tong P, Brown E, Wu P, et al. Cambrian-1: A fully open, vision-centric exploration of multimodal llms[J]. Advances in Neural Information Processing Systems, 2024, 37: 87310-87356.
>
> ---
> **Thank you again for helping us improve the paper and hope our response can resolve your concerns! Please let us know if you have any further questions. We will be actively available until the end of rebuttal period. If you feel your concerns are addressed, please consider reevaluating our work. Looking forward to hearing from you :-) !**

---

> > ### Comment · Reviewer_5eUF · 2025-08-04
> >
> > Thank you for your detailed response. The concerns regarding Q1/W1 and Q3/W3 have been addressed. However, a few questions remain. For Q2/W2, could the authors include a comparison with QwenRL? Additionally, to further demonstrate the generalization capability of VisionThink, it would be helpful to incorporate evaluations using extra VLMs, such as InternVL and LLaVA.

---

> > > ### Author Response · Authors · 2025-08-04
> > > **We're glad to know most questions were addressed; Now we reply to the two more questions**
> > >
> > > Dear Reviewer 5eUF,
> > >
> > > Thank you for your kind response! We're glad to hear that most of your concerns have been addressed. **Your professional feedback has truly strengthened our paper.** Below, we respond to your remaining two points:
> > >
> > > ---
> > >
> > > > ## Add Qwen-RL to the comparison
> > >
> > > Thank you for the reminder. We've now added **Qwen-RL**, **Qwen-RL (1/4)**, and **Down-Sample** baselines to the table below, which further highlights the effectiveness of our VisionThink. We will incorporate this updated comparison into the paper. Your kind suggestion helps make the paper more complete and clearer!
> > >
> > > > Note: VisionThink and Qwen-RL (1/4) are trained on the same resolution as the Down-Sample baseline, while Qwen-RL is trained with 2× resolution (same as Vanilla).
> > >
> > > | Task     | Qwen2.5VL-7B | Down-Sample | Qwen-RL | Qwen-RL (1/4) | VisionThink |
> > > | -------- | ------------ | ---------- | ------- | ------------- | ----------- |
> > > | Counting | 63.1         | 58.9       |  67.1  | 61.9         | **65.7**    |
> > >
> > > ---
> > >
> > > > ## Suggestions for Adding VisionThink to Additional VLMs
> > >
> > > We greatly appreciate your thoughtful suggestion! We also hope to train VisionThink on more backbone models in the future to better validate its generalization capability and further benefit the community.
> > >
> > > However, there are **practical constraints** that make this difficult to achieve within the **limited 3-day discussion window**:
> > >
> > > ### 1. Infrastructure limitations
> > >
> > > We currently build on the **Verl** codebase (11.8k+ stars), which is one of the most powerful and widely used repositories for RL-based training. However, it is also highly complex (e.g., Ray-based code) and **currently only supports the Qwen family**.
> > >
> > >
> > > * For **InternVL**, the most recent RL-relevant implementation is PR #2327(github/Verl/pull/2327) in the Verl GitHub repo, which remains unmerged and still contains some bugs as of three days ago.
> > >
> > > * For **LLaVA**, due to the backbone's limited capability, RL training is **rarely explored** in this line of work, and **Verl does not support LLaVA either**.
> > >
> > > ### 2. Field-wide constraints
> > >
> > > Recent related works [1–5], many of which have received **over 100 citations** within 1–2 months, focus exclusively on Qwen-based models, reflecting the same infrastructure constraints across the field.
> > >
> > >
> > >
> > > ---
> > >
> > > That said, we **fully agree** that this **should not limit future generalization studies** of VisionThink. And we also sincely hope that VisionThink can be validated on a **wider range of models** to encourage broader community adoption.
> > >
> > > Although addressing these engineering challenges is difficult within the discussion phase, **we commit to the following in the camera-ready version:**
> > >
> > > * Training VisionThink on additional models
> > > * Open-sourcing related code and infrastructure
> > >
> > > We sincerely thank you for your understanding and forward-looking suggestions.
> > >
> > > ---
> > >
> > > We truly appreciate your insightful questions and thoughtful discussions, which have significantly improved the quality of our paper.
> > > We will incorporate all new results and analysis into the camera-ready version.
> > >
> > >
> > > **If there are any remaining concerns, we would be happy to address them. If you feel your concerns have been fully addressed, we would greatly appreciate your reconsideration of the overall evaluation**.
> > >
> > > Warm thanks again!
> > >
> > > ---
> > >
> > >
> > >
> > > **References**
> > >
> > > [1] Visual-RFT: Visual Reinforcement Fine-Tuning
> > >
> > > [2] R1-OneVision: Advancing Generalized Multimodal Reasoning through Cross-Modal Formalization
> > >
> > > [3] R1-V: Reinforcing Super Generalization Ability in Vision-Language Models with Less Than \$3
> > >
> > > [4] Search-R1: Training LLMs to Reason and Leverage Search Engines with Reinforcement Learning
> > >
> > > [5] ReTool: Reinforcement Learning for Strategic Tool Use in LLMs

---

> > > > ### Comment · Area_Chair_XGCp · 2025-08-05
> > > >
> > > > As the author-reviewer discussion will end on Aug 8, it would be great to leave your further comments for the authors' reply, or confirm the Mandatory Acknowledgement with your rating.

---

> > > > ### Comment · Reviewer_5eUF · 2025-08-05
> > > > **Thank you for the detailed response!**
> > > >
> > > > The authors have addressed all of my concerns, so I will raise my score to 4. However, I still have a lingering worry that VisionThink relies solely on Qwen and lacks experiments on other MLLMs, which makes it difficult to demonstrate sufficient generalizability and robustness at this stage.

---

> > > > > ### Author Response · Authors · 2025-08-08
> > > > > **Adding VisionThink to Additional VLMs**
> > > > >
> > > > > Dear Reviewer 5eUF,
> > > > >
> > > > > Thank you so much for your warm and encouraging response! We're glad to hear that all of your concerns have been addressed.
> > > > >
> > > > > Over the past few days, we have been actively working to further validate the **generalization capability** of the VisionThink training framework (VERL) by adapting it to  **other VLM backbones**.
> > > > >
> > > > > To this end, we selected the **recently released MiMo-VL-SFT** (June 2025) [6] as the target model, given its strong performance across various benchmarks—especially on **OCR-related tasks**.
> > > > >
> > > > >
> > > > > However, since MiMo-VL’s output format is highly diverse (e.g., the base model also includes <think> tokens), we encountered challenges not only in adapting it for training under VERL, but also during evaluation—particularly in applying the original lmms-eval matching algorithm.
> > > > >
> > > > >
> > > > > Due to the time constraint, we were only able to cleanly evaluate three tasks. We appreciate your understanding and hope that the results below still provide **evidence of generalization**. We will continue improving this aspect in the camera-ready version.
> > > > >
> > > > >
> > > > > | MiMo-VL | ChartQA | OCRBench | MME  |
> > > > > | --------------- | ------- | -------- | ---- |
> > > > > | Vanilla         | 91.3    | 86.6     | 2330 |
> > > > > | Down-Sample     | 69.8    | 73.1     | 2300 |
> > > > > | **VisionThink** (Mimo) | 88.7    | 86.5     | 2326 |
> > > > >
> > > > > ---
> > > > >
> > > > > Thank you again for your thoughtful feedback. It has greatly helped us strengthen our paper, and we will incorporate both this discussion and the additional results into the revised version of the paper.
> > > > >
> > > > > ---
> > > > > **References**
> > > > >
> > > > > [6] MiMo-VL Technical Report

---

> > > > > > ### Comment · Reviewer_5eUF · 2025-08-09
> > > > > > **Thanks**
> > > > > >
> > > > > > Thanks for the response. I suggest the authors integrate these discussions into the final version.

---

> > > > > > > ### Author Response · Authors · 2025-08-09
> > > > > > > **Appreciation**
> > > > > > >
> > > > > > > Thank you for your kind words. We will improve our final version of the paper based on the discussion. We truly appreciate the effort and time you have devoted during both the review and discussion phases. The discussion has also been insightful for us and has led to new findings. Thank you again.

---

### Note · Authors · 2025-08-12

Dear Senior Area Chairs, Area Chairs, and Reviewers,

We hope this letter finds you well.

We sincerely **thank you for your time and effort** throughout the NeurIPS review process. We were **fortunate to have five responsible reviewers** whose in-depth discussions greatly improved our work. We also **thank the AC and SAC for your support during the process** and for **assigning such professional reviewers** — all of whom reported confidence scores ≥4.



**Main Discussions**

* **5eUF**: Discussed perception tasks and generalization.
* **wk5x**: Discussed LLM-as-Judge bias and VisionThink trained on rule-based rewards.
* **2Thu**: Suggested additional fine-grained benchmarks and presentation improvements.
* **maK1**: Discussed decoding time and offered impactful writing suggestions.
* **Ubhv**: Analyzed the effect of input size and penalty design on model behavior.

We thank all reviewers for their insightful suggestions and are pleased that **almost all concerns have been addressed**. As authors cannot view new feedback after Aug 8, any further new points will be discussed in the camera-ready version.


**Broader Impact & Outlook**

* With the rapid development of reasoning models, we encourage the community to pay greater attention to the emerging field of **efficient reasoning VLMs**. We hope that reasoning VLMs will **move beyond narrow domains** such as visual math and **strengthen their general reasoning capabilities**.

* VisionThink explores **a new paradigm for image understanding**, distinct from traditional VLMs. It leverages reinforcement learning to train models to “think with images” and decide whether higher-resolution input is needed, enabling maximum efficiency while preserving performance. This framework **naturally extends to tasks such as image zoom-in and video token selection, offering a smarter and more adaptive approach to visual input processing.**

* VisionThink explores learned decision-making over when and how to use visual resources (e.g., resize), aligning with the broader vision of **VLM-as-agent** settings. We hope our penalty-based mechanism could also offer a little insight for future work on **tool use in multimodal systems**.


**We will open-source our data, code, and models to support the community and contribute to continued progress in this field.**


Lastly, we thank again for your constructive feedback and support throughout the process.


Best regards,

Authors of #4914

---

### Decision · Program_Chairs · 2025-09-17

**Decision:**

Accept (poster)

**Comment:**

This paper introduces a vision-language model that adaptively adjusts the number of visual tokens it processes. The model initially works with a low-resolution image and dynamically utilizes higher-resolution tokens only when it determines they are necessary. This on-demand approach offers a practical solution to the trade-off between computational efficiency and accuracy.

The review process for this paper was highly valuable, marked by thorough discussion and constructive engagement between the authors and all reviewers. This collaborative effort has strengthened the paper and led to a unanimous recommendation for acceptance. The authors' commitment to open-sourcing their data, code, and models is a commendable contribution that will undoubtedly benefit future research in this area. We look forward to seeing the final version of the paper, which is expected to incorporate the valuable points raised during the discussion phase.